# SENP8 limits aberrant neddylation of NEDD8 pathway components to promote cullin-RING ubiquitin ligase function

**Kate E Coleman[1†], Miklós Békés[1†‡], Jessica R Chapman[2], Sarah B Crist[1], Mathew JK Jones[3], Beatrix M Ueberheide[1,2], Tony T Huang[1*]**

[1]Department of Biochemistry and Molecular Pharmacology, New York University School of Medicine, New York, United States; [2]Proteomics Laboratory, Division of Advanced Research Technologies, New York University School of Medicine, New York, United States; [3]Molecular Biology Program, Sloan Kettering Institute, Memorial Sloan Kettering Cancer Center, New York, Unites States

**Abstract** NEDD8 is a ubiquitin-like modifier most well-studied for its role in activating the largest family of ubiquitin E3 ligases, the cullin-RING ligases (CRLs). While many non-cullin neddylation substrates have been proposed over the years, validation of true NEDD8 targets has been challenging, as overexpression of exogenous NEDD8 can trigger NEDD8 conjugation through the ubiquitylation machinery. Here, we developed a deconjugation-resistant form of NEDD8 to stabilize the neddylated form of cullins and other non-cullin substrates. Using this strategy, we identified Ubc12, a NEDD8-specific E2 conjugating enzyme, as a substrate for auto-neddylation. Furthermore, we characterized SENP8/DEN1 as the protease that counteracts Ubc12 auto-neddylation, and observed aberrant neddylation of Ubc12 and other NEDD8 conjugation pathway components in SENP8-deficient cells. Importantly, loss of SENP8 function contributes to accumulation of CRL substrates and defective cell cycle progression. Thus, our study highlights the importance of SENP8 in maintaining proper neddylation levels for CRL-dependent proteostasis.

*For correspondence: tony.huang@nyumc.org

†These authors contributed equally to this work

Present address: ‡Nurix Inc, San Francisco, United States

**Competing interests:** The authors declare that no competing interests exist.

## Introduction

Conjugation of the ubiquitin-like (Ubl) protein NEDD8 (neural precursor cell expressed developmentally downregulated protein 8) to target proteins, or neddylation, is a fundamental biological process that controls many key cellular functions, including cell cycle progression, the DNA damage response, and apoptosis. Like ubiquitylation, neddylation is catalyzed through an enzymatic cascade consisting of an E1-activating enzyme complex (NAE1/APPBP1-UBA3), two E2 enzymes (Ubc12/Ube2M and Ube2F) and several E3 enzymes such as *Really Interesting New Gene* (RING)-box protein 1 (Rbx1) and its close homologue Rbx2 (reviewed in [*Soucy et al., 2010*; *Enchev et al., 2014*; *Brown and Jackson, 2015*]). Additional proteins called DCUN1D1-5 (defective in cullin neddylation, domain containing 1–5) have been described in human cells as cofactors for RBX1 and RBX2, which potentiate neddylation activity (*Kim et al., 2008*; *Kurz et al., 2008*; *Monda et al., 2013*). The most well-studied target for NEDD8 conjugation are the cullins (CUL1,−2,−3,−4A,−4B,−5, and −7), which are the scaffold subunits of the largest family of ubiquitin E3 ligases, the cullin-RING ligases (CRLs). NEDD8 transfer from the E2 to a lysine (Lys) residue within the cullin promotes conformational changes that increase the catalytic activity of the CRL, while also blocking the binding of the CRL exchange factor CAND1 (*Duda et al., 2008*; *Saha and Deshaies, 2008*; *Pierce et al., 2013*). Structural changes induced by NEDD8 conjugation to cullin subunits ultimately contribute to the efficient ubiquitylation of downstream CRL substrates and their degradation by the 26S proteasome.

By inducing CRL activation and assembly in this way, NEDD8 ligation to cullins controls a high proportion of ubiquitylation events in cells (~10–20%) (*Soucy et al., 2009*), making this pathway an attractive target for pharmacological manipulation of protein turnover or proteostasis. An inhibitor of the NEDD8 E1 enzyme, MLN4924, has been recently developed and shown to potently and rapidly suppress neddylation in cells, thereby inhibiting CRL activity. Significantly, preclinical studies using MLN4924 have demonstrated its antitumor activity in several human tumor xenografts and hematological malignancies (*Soucy et al., 2009*, *2010*; *Brownell et al., 2010*). While the antitumorigenic activity of MLN4924 has been largely attributed to CRL substrates, it is unknown whether non-cullin targets of neddylation could also play important roles in tumorigenesis.

Although cullins are the primary substrates for NEDD8 conjugation, more recently several non-cullin neddylation targets have been described. Among this list of proteins are p53 (*Xirodimas et al., 2004*), E2F1 (*Loftus et al., 2012*; *Aoki et al., 2013*), ribosomal protein L11 (*Sundqvist et al., 2009*), Smurf1 (*Xie et al., 2014*) and Histone H4 (*Ma et al., 2013b*). In addition, large-scale proteomic surveys have uncovered several other neddylation substrates both within and outside the CRL pathway (*Jones et al., 2008*; *Xirodimas et al., 2008*). However, some of these non-cullin substrates remain to be fully validated as true NEDD8 targets, as these studies relied on over-expression of epitope-tagged NEDD8 constructs as bait. Since NEDD8 conjugation can also occur through the ubiquitylation machinery under these conditions (*Hjerpe et al., 2012*), it can be difficult to assign *bona fide* neddylation substrates in such experiments. Another challenge in the identification of non-cullin neddylation targets is the relatively low abundance and transient nature of NEDD8 modification events in cells, limiting neddylation detection at an endogenous level by proteomic approaches.

Like other protein post-translational modifications (PTMs), neddylation is reversible. COP9 signalosome complex subunit 5 (CSN5), a metallo-protease and component of the eight-subunit COP9 signalosome complex (CSN), is the major cullin deneddylase in human cells (*Lyapina et al., 2001*; *Cope, 2002*). CSN is specific for neddylated cullins (*Lingaraju et al., 2014*; *Cavadini et al., 2016*); however, deneddylase(s) controlling non-cullin neddylated substrates have been poorly defined (*Figure 1A*). Recently, a cysteine protease called SENP8 (also known as DEN1 or NEDP1) has been characterized that functions distinctly from CSN in deneddylating primarily non-cullin substrates (*Chan et al., 2008*; *Mergner et al., 2015*) as well as hyper-neddylated cullins (*Mendoza et al., 2003*; *Wu et al., 2003*). SENP8 selectively interacts with NEDD8 and not ubiquitin (*Gan-Erdene et al., 2003*; *Shen et al., 2005*), and also plays a redundant role in proteolytic processing of the precursor form of NEDD8 in conjunction with ubiquitin C-terminal hydrolase isozyme 3 (UCHL3) (*Wada et al., 1998*; *Wu et al., 2003*). Criteria defining the unique substrate preferences of CSN and SENP8 are still not clear; however, a previous study showed distinct neddylation defects in DEN1[null] versus CSN5[null] *Drosophila* larvae, suggesting that the two enzymes have non-overlapping functions (*Chan et al., 2008*). Moreover, the specific substrates for NEDD8 deconjugation by SENP8, as well as the phenotypic consequences of long-term SENP8 depletion, have not been thoroughly profiled in mammalian cells.

To capture and enrich for potentially low-level non-cullin neddylated substrates, we employed a strategy to stabilize and trap the 'neddylated' state of proteins by using an ectopically expressed, deconjugation-resistant mutant of NEDD8, as previously performed using uncleavable ubiquitin and SUMO modifiers (*Békés et al., 2011*, *2013*). The deconjugation-resistant form of NEDD8 was expressed in cells using a doxycycline (Dox)-inducible expression system to minimize the potential of overexpression artifacts, followed by immuno-affinity purification and identification of neddylated substrates by mass spectrometry (MS). Strikingly, we observed enrichment of not only neddylated cullins using this technique but also several novel non-cullin targets, including the NEDD8 E2-conjugating enzyme, Ubc12. In a series of experiments, we confirmed that Ubc12 becomes auto-neddylated both in vitro and in cells, and that Ubc12 and other NEDD8 pathway components form aberrant NEDD8-conjugates in the absence of deneddylating activity. Moreover, we identified SENP8, but not CSN, as the deconjugating enzyme responsible for reversing aberrant neddylation of non-cullin substrates. Importantly, SENP8 loss-of-function studies showed that SENP8 is required for proper CRL activation and G1/S phase cell cycle progression. Finally, to identify candidate substrates with altered ubiquitylation status that could contribute to this defective G1/S regulation, we performed an unbiased proteomics screen to compare the relative abundance of ubiquitylated peptides in parental versus SENP8 knockout cells. Using this approach, we uncovered several potentially

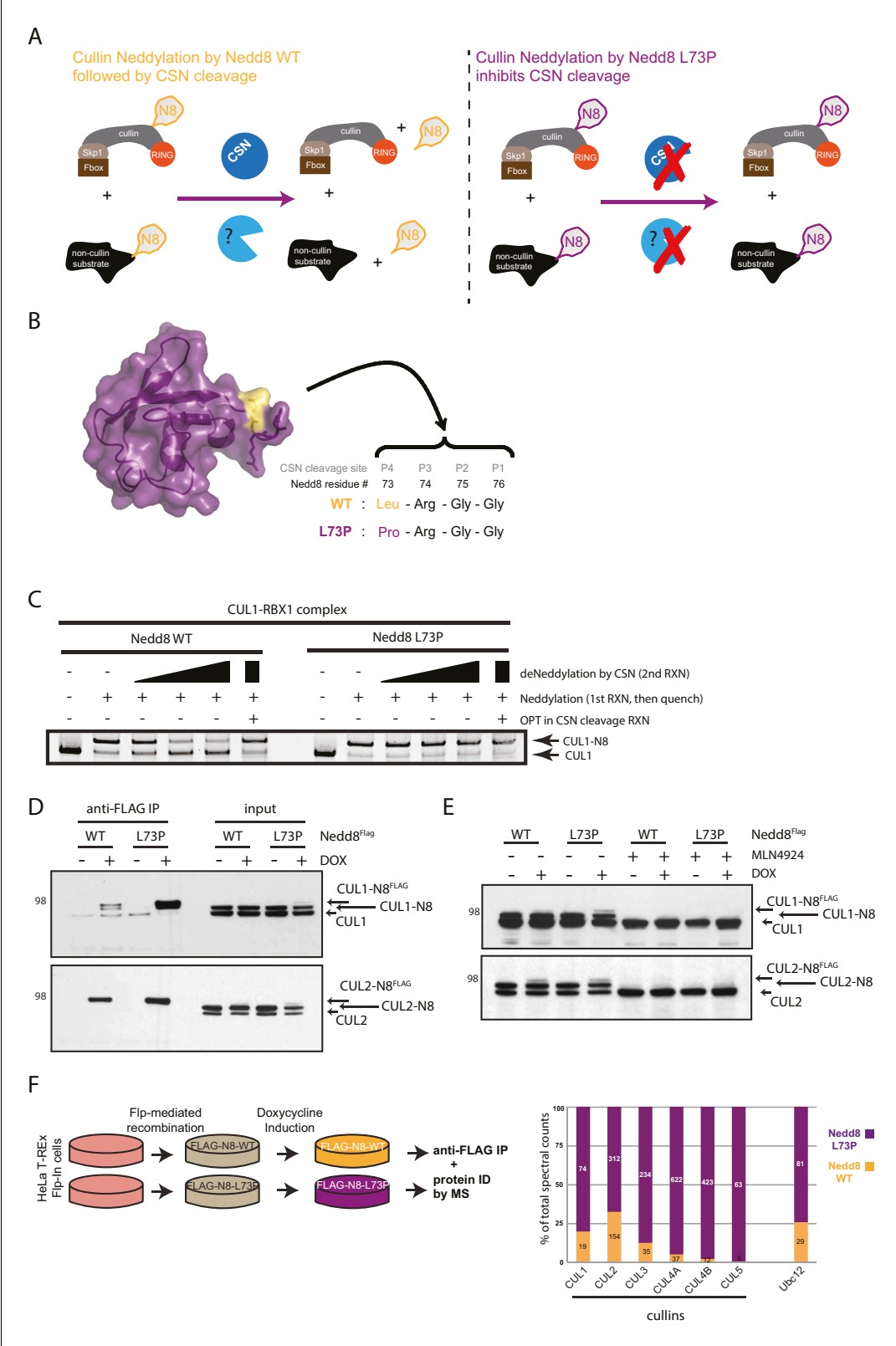

**Figure 1.** Expression of a deconjugation-resistant NEDD8 mutant (L73P) stabilizes neddylation of cullins and other non-cullin substrates. (**A**) Schematics of the regulation of NEDD8 substrates by modification with either WT- (left panel) or L73P-Nedd8 (right panel), and deneddylation by NEDD8-specific proteases. CSN is the deneddylase responsible for deconjugating NEDD8 from cullin substrates, but proteases regulating deneddylation of non-cullin substrates are largely uncharacterized. (**B**) Surface representation of NEDD8 (pdb: 1NDD) and details of its C-terminal tail, showing its proteolytic

*Figure 1 continued on next page*

*Figure 1 continued*

cleavage site and location of the L73P mutation. (C) Recombinant CRL1/Rbx1 was in vitro neddylated by purified His-NEDD8-WT or His-NEDD8-L73P, in the presence of E1 and E2 enzymes and ATP. Reactions were quenched, and recombinant CSN was added at increasing concentrations to monitor the ability of each NEDD8 moiety to be deconjugated from CUL1. OPT (1,10-orthophenatroline, 1 mM) was added to samples containing the highest concentration of CSN (last lane) to completely inhibit CSN activity. (D) FLAG-NEDD8-WT or FLAG-NEDD8-L73P was induced in HeLa-FlpIn-N8 cells using 1 ug/mL doxycycline for 48 hr prior to collection. Whole-cell lysates of untreated or Dox-treated cells were incubated with anti-FLAG beads to purify FLAG-NEDD8-conjugates. Immunoblots of input and IP samples were analyzed for FLAG-NEDD8-modified CUL1 and CUL2. (E) HeLa-FlpIn-N8 cells were treated with or without Dox as in D to induce FLAG-NEDD8-WT or FLAG-NEDD8-L73P, and subsequently incubated with or without the of the CRL inhibitor MLN4924 (5 μM for 4 hr) before harvesting. Whole-cell extracts were analyzed for FLAG-NEDD8-conjugated CUL1 and CUL2. (F) (left panel) Workflow for expression and purification of FLAG-NEDD8-WT and FLAG-NEDD8-L73P for MS analysis. (right panel) Percentages of total spectral counts detected in FLAG-IPs from cells expressing either FLAG-NEDD8-WT (orange bars) or FLAG-NEDD8-L73P (purple bars). The numbers in the columns indicate actual spectral counts. The IPs were performed on lysates from the same number of cells.

The following source data is available for figure 1:

**Source data 1.** NEDD8- modified peptides identified by MS analysis of FLAG-NEDD8 IP samples.

relevant substrates with reduced ubiquitylation and proteasome-mediated degradation in SENP8-deficient cells that have important roles in cell cycle regulation. Therefore, our findings suggest a novel role for SENP8 in the quality control of the NEDD8 conjugation pathway through prevention of aberrant neddylation of pathway components and demonstrate the functional consequences of perturbing this regulation on CRL activation and cell growth control.

# Results

## Identification of novel neddylation substrates in cells by expression of a deconjugation-resistant NEDD8 mutant

We have previously demonstrated that mutation of leucine (Leu) 73 in ubiquitin, located at the P4 position of the DUB cleavage site at the C-terminus, to proline (Pro) renders conjugated ubiquitin resistant to cleavage by a wide range of deubiquitinating enzymes (DUBs) (*Békés et al., 2013*). The C-termini of Nedd8 and Ub are conserved, where the P4 position in NEDD8 is also a Leu residue. Therefore, to generate a deconjugation-resistant form of NEDD8 to be utilized in the identification of non-cullin neddylation substrates, we engineered the analogous mutation in NEDD8 (L73P) (*Figure 1B*). To test whether this mutant form of NEDD8 was indeed refractory to deneddylation, we expressed and purified recombinant His-tagged wild-type (WT) NEDD8 and the L73P NEDD8 mutant and used them to in vitro neddylate recombinant CUL1 (and other cullins, data not shown) followed by CSN-mediated deneddylation (*Figure 1C*). While WT NEDD8 was readily deconjugated from CUL1 with increasing concentrations of CSN in the reaction, the L73P NEDD8 mutant was resistant to cleavage from CUL1 at all CSN concentrations. As expected, the addition of the zinc-chelator ortho-phenanthroline (OPT) to the reaction mixture completely inhibited the metalloprotease activity of CSN, and neither WT nor L73P NEDD8 were cleaved from CUL1 under these conditions (*Figure 1C*). Collectively, these in vitro studies established NEDD8 L73P as a deneddylation-resistant mutant.

Next, we used NEDD8 L73P as a tool to trap neddylated forms of substrates in cells, purify stabilized NEDD8-conjugates, and identify substrates and neddylation sites by MS. For this purpose, we generated Dox-inducible (Flp-In) HeLa cell lines expressing low levels of either FLAG-NEDD8-WT or FLAG-NEDD8-L73P, as had been previously developed for Ub (*Békés et al., 2013*), and used spectral counts to determine the relative abundance of WT- versus L73P-purified NEDD8 substrates. Immunoblot analysis of FLAG-immunoprecipitations (IPs) performed using these cell lines showed that both versions of ectopically expressed NEDD8 could be incorporated into CUL1- and CUL2-NEDD8 conjugates (*Figure 1D*). We observed increased recovery of FLAG-NEDD8-L73P-modified CUL1 compared to FLAG-NEDD8-WT-conjugated CUL1, likely reflecting the inability of the L73P mutant to be deneddylated in cells (*Figure 1D*). Consistent with previous reports (*Soucy et al., 2009*; *Brownell et al., 2010*), MLN4924 treatment blocked the conjugation of both endogenous NEDD8 and FLAG-NEDD8 to cullin substrates entirely (*Figure 1E*). Importantly, immunoblot

analyses showed that both FLAG-NEDD8-WT and FLAG-NEDD8-L73P were incorporated into CUL1- and CUL2- conjugates at similar levels as endogenous NEDD8 (*Figure 1E*). Taken together, these experiments show the utility of both the WT and L73P NEDD8 constructs in isolating neddylated cullins and potentially other non-cullin substrates, in a manner that does not involve the spurious effects of NEDD8 overexpression studies.

We subsequently scaled up and purified both FLAG-NEDD8-WT and FLAG-NEDD8-L73P conjugates in sufficient quantities for MS analysis. (*Figure 1F* and *Figure 1—source data 1*). As anticipated, both FLAG-NEDD8-WT and FLAG-NEDD8-L73P immunoprecipitated CUL1, −2, −4A, and −4B, although the relative abundance, based on spectral counting, of these cullin substrates was much higher in the FLAG-NEDD8-L73P IP samples (*Figure 1F* and *Figure 1—source data 1*). Interestingly, several non-cullin neddylation targets were identified using this strategy (for a complete list, see *Figure 1—source data 1*), most notably the NEDD8 E2 conjugating enzyme Ubc12 (*Figure 1F*) as well as other previously validated substrates such as Histone H4 (*Ma et al., 2013b*) and ribosomal protein L11 (*Sundqvist et al., 2009*). Given the particularly high prevalence of Ubc12 peptide spectral matches (PSMs) in the FLAG-NEDD8-L73P IPs, our results prompted us to further investigate Ubc12 as a potential novel neddylation substrate.

## Validation of Ubc12 as a substrate for neddylation

Upon further analysis of our MS results from the FLAG-NEDD8-L73P IP dataset, we determined that the site of NEDD8 conjugation to Ubc12 is Lys3, as we observed a K-ε-GG remnant, characteristic of trypic digestion of either NEDD8 or ubiquitin, at this site (*Figure 2A*). This Ubc12 peptide was also acetylated at the N-terminus, in agreement with previous observations (*Scott et al., 2011*). Sequence alignments show that the Lys3 residue of Ubc12 is highly conserved across several eukaryotic orthologs (*Figure 2A*). Immunoblot analysis of identically prepared FLAG-NEDD8 IPs showed that Ubc12 is modified upon Dox-induced expression by FLAG-NEDD8-L73P, but not when pretreated with MLN4924, (*Figure 2B*). suggesting that these represent neddylated Ubc12 species (*Figure 2B*). To confirm that these modified forms of Ubc12 resulted from NEDD8 conjugation and not ubiquitylation, we compared relative binding of either ubiquitin L73P or NEDD8 L73P to ectopically expressed Ubc12 in HEK 293T cells. Both WT and L73P HA-tagged NEDD8 constructs, but not the analogous WT and L73P HA-tagged Ub constructs, formed conjugates with co-transfected Myc-Flag-Ubc12 WT in these experiments (*Figure 2C*). We derived additional Ubc12 constructs to test whether the K3 site could support modification by NEDD8 L73P. As we noted that there were several alternative Lys sites at the N-terminus of Ubc12 that could potentially be modified other than K3 (indicated in the alignments in *Figure 2A*), we engineered an additional mutant (NK0) in which all N-terminal Lys residues were converted to Arginine (Arg). As shown in *Figure 2D*, mutations introduced into the NK0-Ubc12 mutant sufficiently blocked modification by co-transfected NEDD8-L73P. Furthermore, restoring a single Lys at position 3 (K3) in the NK0-Ubc12 mutant (N-K3) partially rescued the ability of Ubc12 to form conjugates with NEDD8-L73P (*Figure 2D*). Altogether, these results provide strong evidence that Ubc12 is a *bona fide* substrate for neddylation, and that N-terminal Lys sites, including K3, are critical for NEDD8 modification of Ubc12.

It was previously shown that the acetylated N-terminal Met in Ubc12 interacts with a hydrophobic pocket in the co-E3, DCN1 (DCUN1D1 in human cells), to promote cullin neddylation (*Scott et al., 2011*). Therefore, we sought to determine whether neddylation at the N-terminus of Ubc12 could also impact binding to DCN1 as well as other NEDD8 pathway components. We transiently transfected cells with either Flag-tagged WT or NK0 Ubc12 in the presence of HA-NEDD8-L73P and performed co-IPs in the presence of co-expressed HA-NEDD8-L73P to compare their relative interaction to DCN1 and other factors. Consistent with our previous experiments, only the WT Ubc12 construct, but not the NK0 mutant, could support modification with HA-NEDD8-L73P, which likely contributed to reduced interaction to DCN1 and Cul3 (*Figure 2E*). Consistently, WT Ubc12 co-purified with more DCN1 when it was not neddylated by HA-NEDD8-L73P co-expression (*Figure 2F*). Collectively, these results suggest that N-terminal neddylation of Ubc12 likely weakens binding to DCN1, which in turn may influence downstream NEDD8 pathway function.

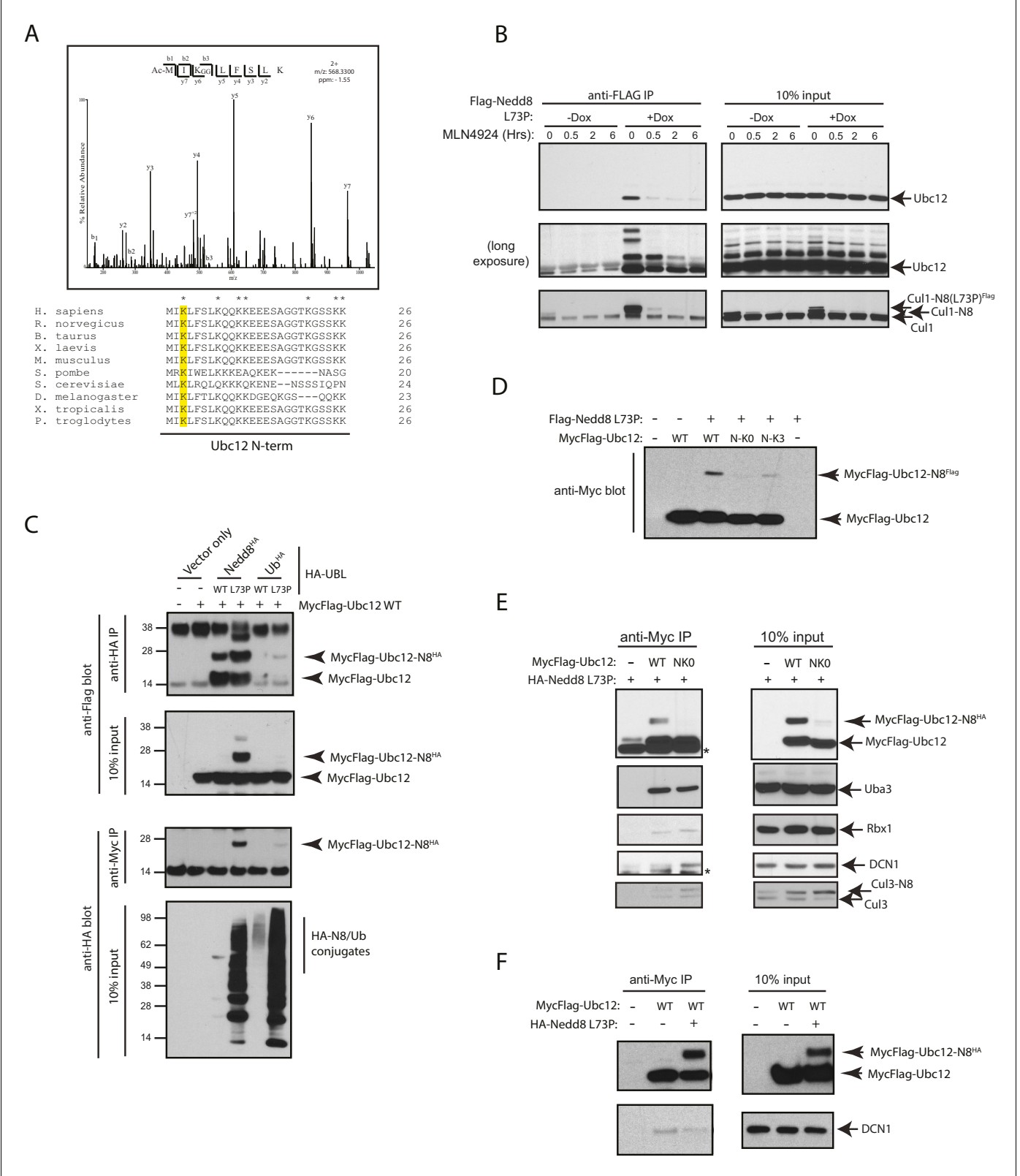

**Figure 2.** Nedd8 L73P is conjugated to Ubc12 in vitro and in cells. (**A**) An HCD MS$^2$ spectrum of the doubly charged Ubc12 N-terminal peptide, MIKLFSLK, N-terminally acetylated and K-ε-GG modified on Lys3 (top) and sequence alignment of Ubc12 N-termini from the indicated species (bottom). The conserved Lys3 residue in the Ubc12 N-terminus identified as a K-ε-GG-modified residue from the MS dataset is highlighted in yellow. Lys residues mutated to Arg in the Ubc12 NK0 mutant are indicated by asterisks. Sequence alignments were generated using Clustal Omega. (**B**) HeLa-

*Figure 2 continued on next page*

*Figure 2 continued*

Flp-In-NEDD8 cells were either left untreated or treated with Dox to induce FLAG-NEDD8-L73P expression, and subsequently treated with MLN4924 (5 µM) for the indicated times. Immunoblots of input and anti-FLAG IP samples were analyzed for modified Ubc12 and CUL1. (C) HA-tagged (WT or L73P) NEDD8 or Ub constructs were co-transfected with MycFlag-Ubc12 in HEK293T cells. Conjugation of either NEDD8 or Ub to Ubc12 was analyzed by reciprocal IPs using either anti-HA or anti-Myc antibodies and immunoblotting. (D) MycFlag-tagged WT, NK0 , or N-K3 (NK0 mutant with K3 restored) Ubc12 constructs were co-transfected with Flag-Nedd8-L73P in HEK293T cells. Modified Ubc12 was analyzed in whole-cell lysates by immunoblotting with anti-Myc antibody. (E) WT or NK0 MycFlag-Ubc12 was co-expressed in HEK293T cells with HA-NEDD8-L73P. Cell lysates were incubated with anti-Myc antibody to IP MycFlag-Ubc12, and input and IP samples were analyzed by immunoblotting with the indicated antibodies. Asterisks signify non-specific bands. (F) WT MycFlag-Ubc12 was transfected into HEK293T cells in the presence or absence of HA-Nedd8-L73P. Anti-Myc antibody was used to IP MycFlag-Ubc12 from cell extracts, and input and IP samples were analyzed by immunoblotting with the indicated antibodies.

## SENP8 deneddylates Ubc12

Based on our observation that Ubc12 neddylation is stabilized by the deconjugation-resistant FLAG-NEDD8-L73P mutant (*Figure 2C*), we predicted the presence of a Ubl protease that selectively deconjugates neddylated Ubc12 in cells. To identify the protease responsible for deneddylating Ubc12, we depleted cells for either of the two known deneddylating enzymes (CSN and SENP8) and examined effects on Ubc12 neddylation. Treatment of cells with siRNAs targeting SENP8, but not CSN5, resulted in increased levels of Ubc12 conjugates (*Figure 3A*). These results were corroborated by CRISPR/Cas9-mediated knockout of endogenous SENP8 using multiple sgRNA sequences in different cell lines (*Figures 3B* and *5B*). To confirm that these modified forms of Ubc12 are NEDD8-conjugated, we additionally performed IPs for endogenous Ubc12 in WT and SENP8 knockout HEK293T cells. In immunoblot analyses of IP samples, probing with antibodies against Ubc12 and NEDD8 detected a band of similar molecular weight corresponding to di-neddylated Ubc12 in both SENP8 knockout cell lines, but not the parental cells (*Figure 3—figure supplement 1A*). Additionally, higher molecular weight Ubc12 NEDD8-conjugates were no longer observed in either siRNA-depleted (*Figure 3—figure supplement 1B*) or CRISPR-generated SENP8-deficient cells (*Figure 3—figure supplement 1C*) following treatment with MLN4924. To determine whether SENP8 can directly deconjugate neddylated Ubc12, we utilized an in vitro deneddylation assay using purified components. As demonstrated previously (*Huang et al., 2005*), we observed that Ubc12 undergoes auto-neddylation when incubated with the E1 enzyme, ATP, and recombinant NEDD8 (WT or L73P) (*Figure 3C*). We also showed that Ubc12 auto-neddylation occurs in cis via its own catalytic Cys residue, as the catalytically inactive mutant of Ubc12 (C111A) is not neddylated in vitro (*Figure 3—figure supplement 2A*), even in the presence of additional WT Ubc12 or when transfected in cells (*Figure 3—figure supplement 2B*). This modification was removed when recombinant SENP8 was added to reactions containing WT NEDD8, but not the L73P mutant (*Figure 3C*). Moreover, in titration experiments using different concentrations of recombinant SENP8 or CSN proteases, we further showed that NEDD8 deconjugation from auto-neddylated Ubc12 only occurs in the presence of SENP8, but not CSN (*Figure 3D*). Taken together, these findings establish SENP8 as the primary protease that regulates Ubc12 auto-neddylation both in vitro and in cells.

## Proteomic analysis reveals aberrantly neddylated substrates in SENP8 knockout cells

We considered the possibility that SENP8 may play a role in restricting neddylation of other NEDD8 conjugation pathway regulators in addition to Ubc12. Supporting this notion, immunoblotting analysis of whole-cell extracts with an anti-NEDD8 antibody revealed a dramatic increase in protein neddylation in SENP8 knockout cells in comparison to control cells (*Figure 4A*). This suggests that SENP8 is likely to restrict aberrant or hyper-neddylation of non-cullin substrates in normal steady-state conditions.

To identify novel substrates for deneddylation by SENP8, we utilized an unbiased proteomic approachto quantify differences in NEDD8-derived di-Glycine (K-ε-GG) remnants on trypsinized peptides from WT or SENP8 knockout cell lysates. For this purpose, we employed a previously developed strategy that involves antibody-based enrichment of peptides with the K-ε-GG-remnant produced by tryptic digestion of proteins modified by NEDD8, ubiquitin, or the ubiquitin-like modifier ISG15 (*Xu et al., 2010*; *Emanuele et al., 2011*; *Kim et al., 2011*; *Udeshi et al., 2013*). As shown

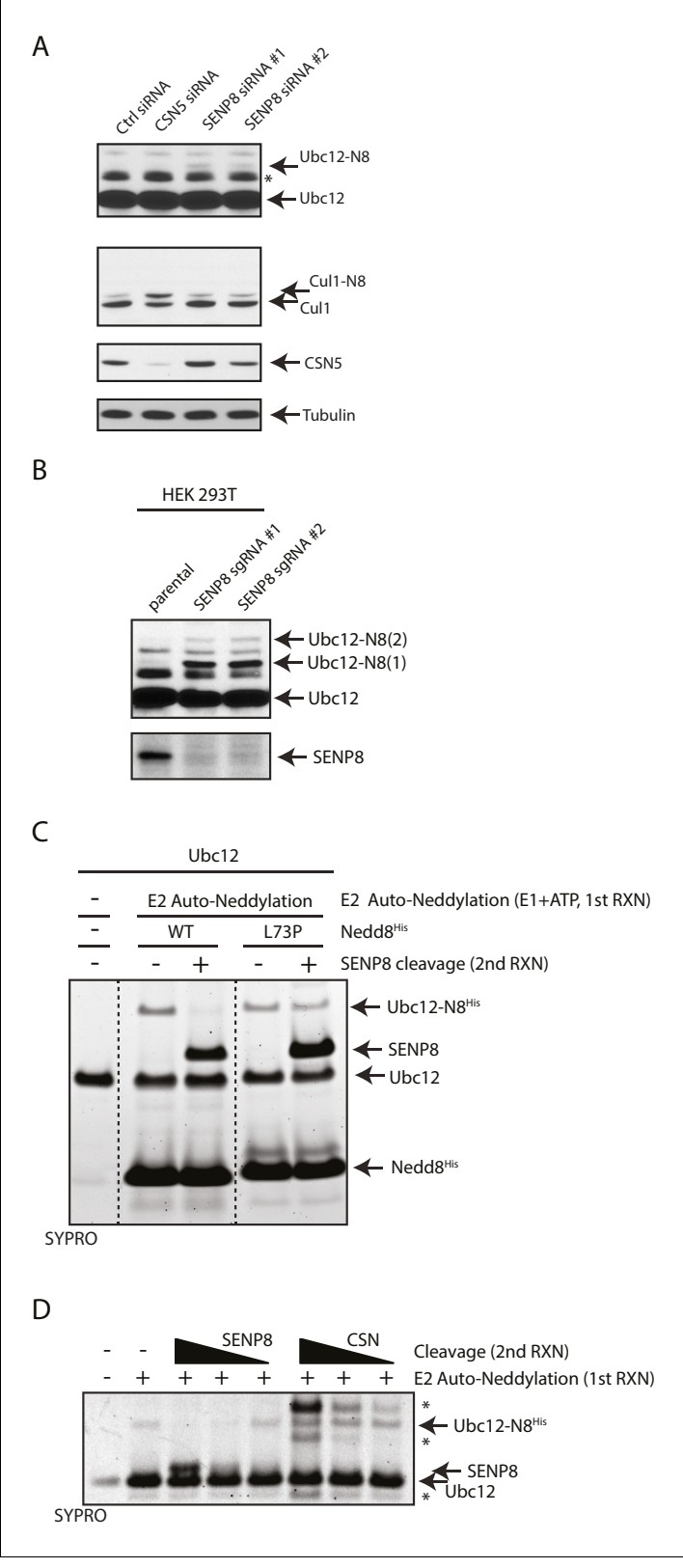

**Figure 3.** SENP8 regulates Ubc12 deneddylation. (**A**) HeLa cells were treated with control, CSN5, or SENP8 siRNAs for 48 hr prior to harvesting. Whole-cell lysates were analyzed by immunoblotting for the indicated proteins. (**B**) SENP8 knockout clones (#1 and #2) of HEK293T cells were derived as specified in Materials and methods. Cell lysates were analyzed by immunoblotting with the indicated antibodies. (**C**) Recombinant Ubc12
*Figure 3 continued on next page*

*Figure 3 continued*

was auto-neddylated in vitro in the presence of E1 enzyme, ATP, and either WT or L73P His-tagged NEDD8. Reactions were quenched, and SENP8 was added to deneddylate Ubc12-NEDD8 conjugates. Deneddylation of recombinant Ubc12 was analyzed by SDS-PAGE and SYPRO staining. (**D**) Ubc12 was in vitro neddylated as in (**C**) in the presence of WT His-tagged NEDD8. Decreasing concentrations of either recombinant SENP8 or CSN were added to quenched reactions, and Ubc12 deneddylation was assessed by SDS-PAGE and SYPRO staining.

The following figure supplements are available for figure 3:

**Figure supplement 1.** Detection of NEDD8-modified Ubc12 in SENP8-deficient cells.

**Figure supplement 2.** Modification of Ubc12 with NEDD8-L73P is dependent on Ubc12 catalytic activity.

---

in the workflow schematic in *Figure 4B*, WT and SENP8 knockout HEK293T cells were alternatively labeled with light and heavy isotopes using the stable isotope labeling by amino acids in cell culture (SILAC) method (*Ong and Mann, 2006*), and subsequently lysed and mixed together at a 1:1 ratio based on protein concentration. The mixed lysate was then subjected to tryptic digestion and antibody-based enrichment for K-ε-GG remnants prior to MS analysis. Two biological replicates are plotted in in *Figure 4C* comparing the SILAC ratios of all K-ε-GG remnant peptides identified from two alternative labeling scheme experiments (forward and reverse SILAC) (see also *Figure 4—source data 1*). We quantified 777 K-ε-GG remnant peptides that were identified in both replicates (*Figure 4—figure supplement 1*).

Our quantitative analysis revealed large increases in the relative abundance of several individual K-ε-GG-modified peptides in SENP8 knockout cells , highlighted in *Figure 4C and D*. Interestingly, many of these K-ε-GG modification changes were observed for proteins involved in the NEDD8 conjugation pathway, including not only Ubc12, but also Ube1C, DCUN1D5, and NEDD8 itself (K11 and K48). We also observed a significant decrease in the NEDD8 modification of CUL5in the SENP8-deficient cell line (*Figure 4C and D*). Overall, these results provide further evidence to suggest that SENP8 prevents aberrant or hyper-neddylation on a global level of multiple proteins within the NEDD8 conjugation network.

Next, we determined whether NEDD8 conjugation to the NEDD8 pathway components identified by MS could be observed in SENP8-deficient cells by immunoblotting (*Figure 5A*). Using lysates from two different cell lines (HEK293T and HeLa) in which SENP8 knockout clones were generated using multiple sgRNA sequences, we monitored changes in NEDD8 conjugation levels in target substrates based on the predicted mass shift for NEDD8 modification (9 kDa). We observed evidence for increased levels of NEDD8 conjugates, and in several cases even di-neddylated ($N8^2$) species, of several NEDD8 pathway constituents in all SENP8-deficient cell lines tested, including NAE1 (APPBP1)-UBA3, Ubc12, and DCUN1D1 and DCUN1D5 (*Figure 5B*). In agreement with our analyses in *Figure 2*, we also showed that relative binding of endogenous Ubc12 to DCN1 was reduced in the SENP8-deficient background in comparison to SENP8-corrected cells, suggesting that the aberrant neddylation observed in SENP8 knockout cells may contribute to a weakened Ubc12-DCN1 interaction (*Figure 5—figure supplement 1A*).

Intriguingly, we also observed a decrease in total levels of the E2 conjugating enzyme UBE2F in the SENP8 knockout cell lines (*Figure 5B*). A previous study demonstrated that while Ubc12 targets all other cullin subunits for activation, UBE2F has a unique preference for CUL5 neddylation (*Huang et al., 2009a*). Thus, the decreased abundance of UBE2F in the SENP8 knockout cell lines could potentially explain the decreased presence of K-ε-GG-modified CUL5 peptides in our quantitative MS experiments (*Figure 4C and D*). To examine cullin neddylation in closer detail, we performed immunoblotting analysis for CUL1, −2,–3, −4A, and −5 in WT and SENP8 knockout clones. Consistently, we observed a marked decrease in CUL5 as well as CUL1 neddylation in SENP8-deficient cells, whereas the neddylation of other cullin subunits was not substantially impacted (*Figure 5C*). Importantly, the neddylation defects observed in SENP8 knockout cells were reversed by complementation with ectopically expressed WT SENP8 (*Figure 5D*). We also observed a restoration of CUL1 and CUL5 neddylation levels when we overexpressed WT Ubc12 and DCN1 constructs in SENP8-deficient cells, further suggesting that loss of SENP8 affects optimal Ubc12/DCN1 function

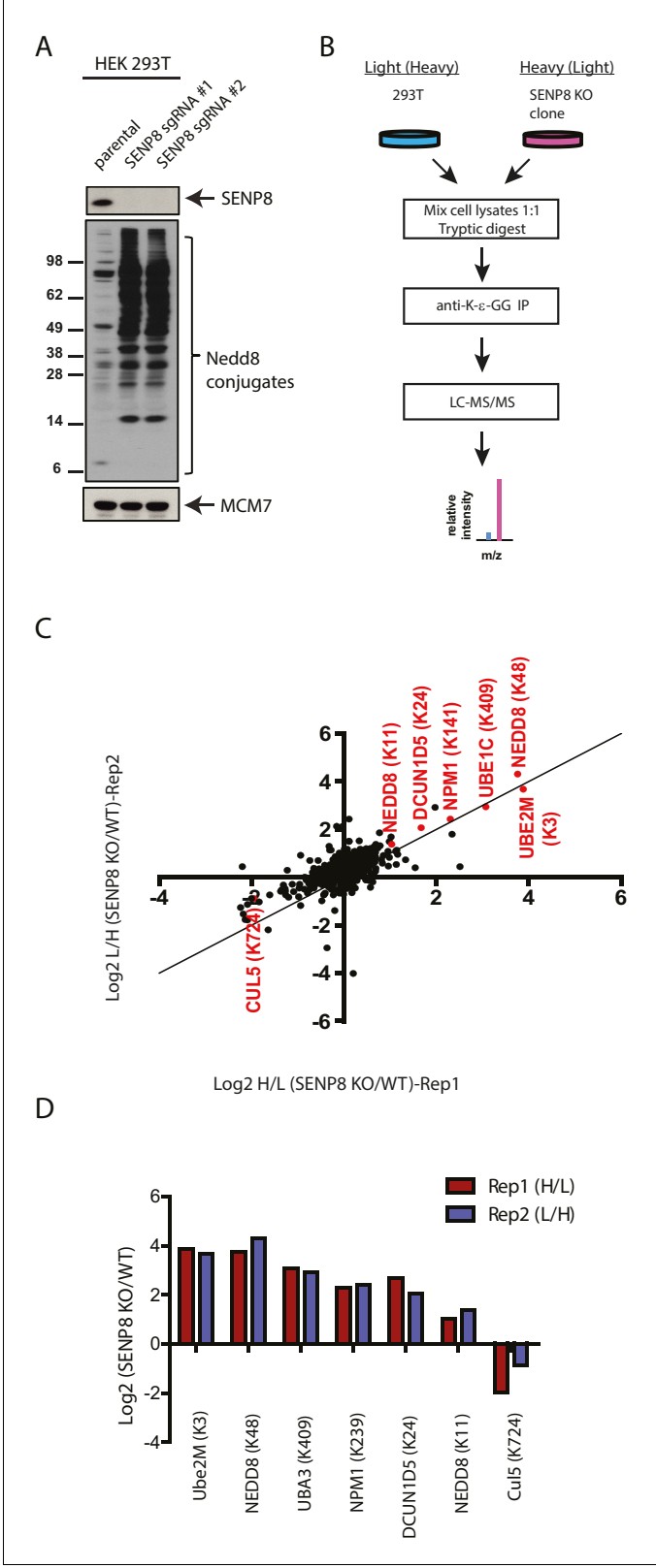

**Figure 4.** Identification of endogenous neddylation substrates in SENP8-deficient cells. (**A**) Lysates from parental and CRISPR-generated SENP8 knockout HEK293T cells were immunoblotted for the indicated proteins. For this figure and all subsequent figures, MCM7 serves as a loading control. (**B**) Workflow for identification of K-ε-GG-remnant containing peptides in WT and SENP8 knockout cells by antibody-based enrichment and quantitative

*Figure 4 continued on next page*

*Figure 4 continued*

SILAC mass spectrometry. (**C**) Scatter plot showing Log2 ratios of all K-ε-GG sites identified in both the forward and reverse SENP8 KO:WT SILAC samples, prepared as in B. K-ε-GG sites of interest identified in both replicates are highlighted in red. Line of identity with slope = 1 is shown for reference. (**D**) Log2 ratios of K-ε-GG modification sites highlighted in C. Error bars represent SEM.

The following source data and figure supplement are available for figure 4:

**Source data 1.** SILAC analysis of K-ε-GG remnant-containing peptides detected in untreated parental and SENP8 knockout cell lysates.
**Figure supplement 1.** Statistical information related to K-ε-GG MS screen comparing untreated parental and SENP8 knockout cell lysates (*Figure 4B*).

(*Figure 5—figure supplement 1B*). These combined quantitative MS and immunoblotting analyses strongly indicate that SENP8 depletion contributes to a variety of neddylation abnormalities in cells.

## SENP8 depletion contributes to decreased CRL activity and altered cell cycle progression

Based on the decreased CUL1 and CUL5 neddylation upon loss of SENP8, we investigated the functional consequences of SENP8 depletion on the degradation of downstream CRL substrates. As several known CUL1 substrates play critical roles in controlling cell proliferation, we first analyzed the effects of SENP8 loss on cell cycle distribution, which can be monitored through flow cytometry analysis of DNA content. For these studies, we used SENP8-deficient and SENP8 WT-corrected HeLa cells, generated as described in Materials and methods (*Figure 6A*). Notably, we observed that the SENP8-deficient cells showed accelerated cell growth in comparison to parental HeLa cells (*Figure 6B*). To determine whether alterations in cell cycle distribution could account for this difference in overall growth rate, we performed analysis of DNA synthesis (measured by EdU incorporation) versus DNA content (measured by DAPI staining) by flow cytometry. Strikingly, we found that loss of SENP8 contributes to notable changes in the cell cycle profiles of asynchronously dividing cells, including a reduction in the subpopulation of G1 cells, and a concomitant increase of cells in S and G2/M phases (*Figure 6C and D*). Importantly, the reduction in G1 phase cells observed in SENP8-deficient cells was partially rescued through complementation with WT SENP8 (*Figure 6C and D*).

To determine whether SENP8 loss affects the timing with which cells progress through the G1/S transition, we additionally synchronized cells in prometaphase by sequential thymidine and nocodozole blocks, and released cells into fresh media for times corresponding to G1 (2–6 hr) or S phase (6–10 hr). Entry into S phase was measured by pulse-labeling cells with EdU prior to collection (a marker for nascent DNA synthesis) and flow cytometry analysis. Interestingly, while EdU incorporation normally begins at 6–8 hr following release from nocodozole in the parental HeLa cell line, the accumulation of EdU positive cells occurs earlier at 4–6 hr in SENP8 knockout cells (*Figure 6E*). This premature EdU incorporation could be partially reversed by complementation with wild-type SENP8 (*Figure 6E*). Collectively, these results show that SENP8 function is important in maintaining proper cell growth and G1/S cell cycle progression.

## SENP8-deficient cells show altered protein ubiquitylation levels and increased stability of key substrates involved in cell cycle regulation

As our results indicate reduced cullin neddylation and function in SENP8-deficient cells, we next sought to identify relevant CRL substrates with altered ubiquitylation and stability in these cells that could potentially contribute to the observed cell growth and proliferation defects. To identify and characterize these proteins in an unbiased manner, we again utilized the SILAC-MS and K-ε-GG remnant peptide enrichment strategy as in *Figure 4B*, but this time both WT and SENP8-deficient HEK293T cells were treated with MG132 prior to harvesting to block proteasome-mediated degradation of ubiquitylated substrates (*Figure 7A*). Since K-ε-GG remnants are also produced following tryptic digestion of ubiquitylated proteins, K-ε-GG profiling and SILAC-MS analysis of MG132-

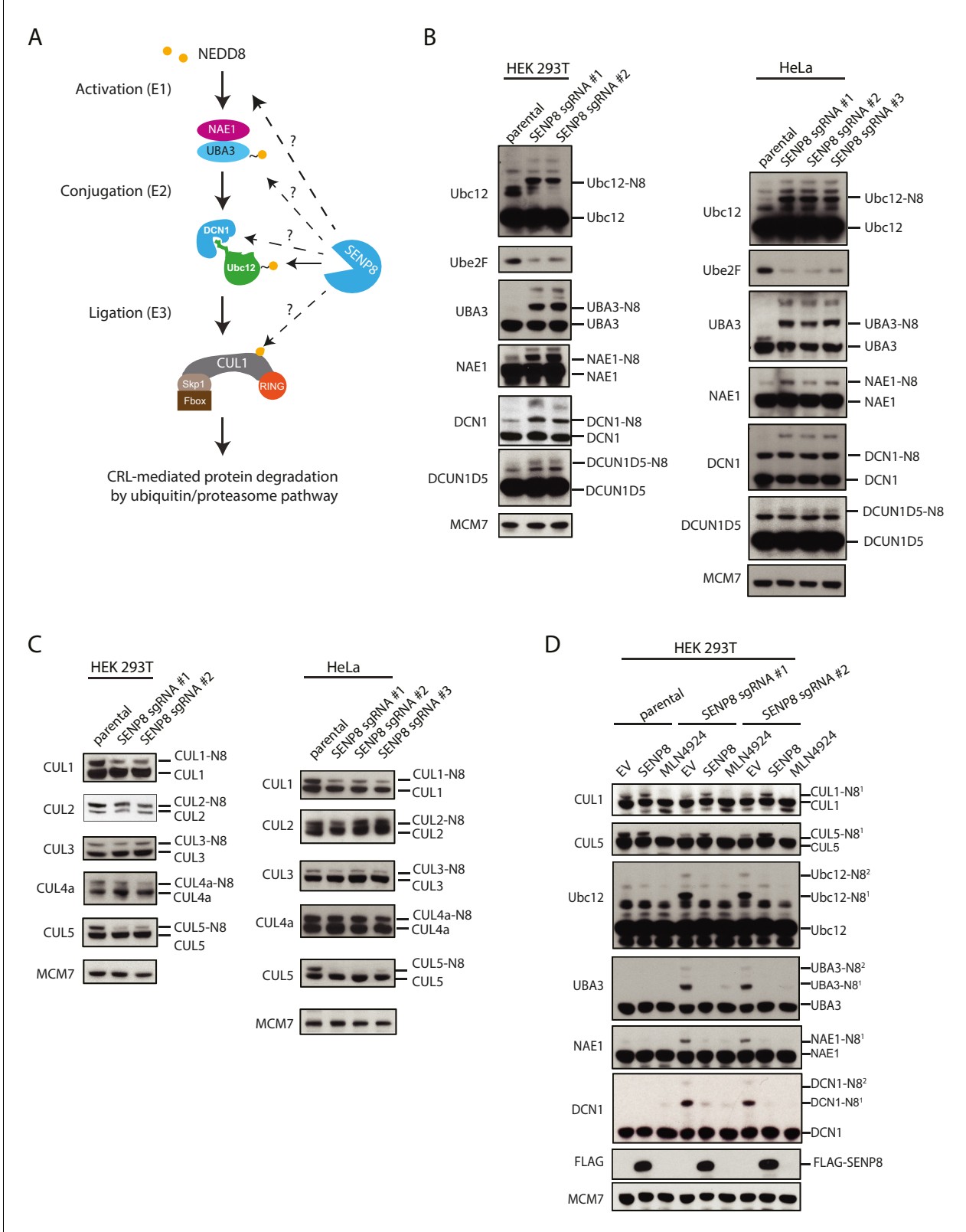

**Figure 5.** Loss of SENP8 leads to aberrant neddylation of NEDD8 pathway components. (**A**) Schematic of the NEDD8 conjugation pathway. Dotted arrows represent potential points of regulation by SENP8-mediated deneddylation. Orange circle represents NEDD8. (**B and C**) Cellular extracts from parental and CRISPR-generated SENP8 knockout HEK293T (left) and HeLa (right) cells were subjected to immunoblot analysis for (**B**) NEDD8 pathway components and (**C**) cullin subunits. NEDD8-conjugated substrates are indicated. MCM7 serves as a loading control. (**D**) Parental or SENP8 knockout

*Figure 5 continued on next page*

Figure 5 continued

HEK293T cells were transfected with empty vector (EV) or a FLAG-SENP8-WT construct and analyzed for NEDD8-conjugated proteins by immunoblotting. As a control, samples from each cell line were treated with 0.3 µM MLN4924 for 24 hr to completely inhibit neddylation of the indicated proteins.
The following figure supplement is available for figure 5:

**Figure supplement 1.** Correction of DCN1 binding and cullin neddylation defects in SENP8-deficient cells.

treated cells should allow for the quantification of differentially ubiquitylated substrates in addition to neddylation targets in SENP8 knockout cells. The SILAC-MS analysis was done using two biological replicates (*Figure 7B* and *Figure 7—figure supplement 1*). Interestingly, K-ε-GG-modified peptides from proteins involved in cell proliferation were relatively more abundant in WT cells versus SENP8 knockout cells (*Figure 7C* and *Figure 7—figure supplement 1D*), indicative of enhanced ubiquitylation of these targets. Importantly, these same K-ε-GG peptides were either completely undetected or of low abundance in the untreated SILAC datasets from *Figure 4B*, as would be expected for MG132-sensitive substrates (*Figure 7C*, see also *Figure 7—source data 1*). Notable targets with reduced ubiquitylation status in SENP8 knockout cells include substrates of the Cul1-based Skp1/Cul1/F-box protein (SCF)$^{\beta TRCP}$ E3 ubiquitin ligase complex, such as Cdc25A (*Donzelli et al., 2002*; *Busino et al., 2003*; *Jin et al., 2003*), UHRF1 (ICBP90) (*Chen et al., 2013*), and β-catenin (*Latres et al., 1999*; *Walker et al., 2015*), as well as other proteins identified as CRL substrates in previously published screens (*Yen and Elledge, 2008*; *Emanuele et al., 2011*).

To further explore whether the stability of relevant CRL substrates is impacted by SENP8 loss, we used cyclohexmide treatment to block de novo protein synthesis (*Figure 7D* and *Figure 7—figure supplement 2A*) and compared total protein levels in asynchronous WT and SENP8 knockout cells by immunoblotting (*Figure 7E*). SENP8 deficiency results in the prolonged half-life of Cdc25A in cycloheximide chase assays, a protein which normally exhibits very rapid turnover in human cells (*Busino et al., 2003*) (*Figure 7D*). SENP8 loss also contributes to the reduced accumulation of higher molecular weight species of Cdc25A following MG132 treatment, indicative of decreased Cdc25A polyubiquitylation in comparison to WT cells. (*Figure 7—figure supplement 2B*). Furthermore, asynchronous SENP8-deficient cells show increased total levels of proteins identified by our MS screen, as well as other key substrates of the SCF$^{Skp2}$, CRL4$^{Cdt2}$, and SCF$^{Fbw7}$ E3 ubiquitin ligases with involvement in DNA replication and cell cycle control. These proteins include the origin licensing factor Cdt1 (*Li et al., 2003*; *Jin et al., 2006*; *Nishitani et al., 2006*), the histone H4 lysine 20 monomethylase Set8 (PR-Set7) (*Abbas et al., 2010*; *Centore et al., 2010*; *Wu et al., 2010*), the cyclin-dependent kinase (CDK) inhibitor p21 (*Bornstein et al., 2003*; *Abbas et al., 2008*) (*Figure 7E*) and the regulatory cyclin for CDK2, Cyclin E1 (*Koepp et al., 2001*; *Hao et al., 2007*).

Considering the cell cycle regulatory roles of some of the ubiquitylated substrates identified in our SILAC-MS screen above (see Discussion for more details), we additionally monitored substrate levels in synchronized cells to determine downstream consequences on G1/S phase progression. For this purpose, we performed the same nocodozole-release experiment as in *Figure 6E* in parental and SENP8-deficient cells, and analyzed relative levels of cyclins and other G1/S regulatory proteins by immunoblotting. Notably, the characteristic G1/S degradation patterns of both Cdc25A and UHRF1, identified from our SILAC-MS screen in *Figure 7B*, were substantially altered in SENP8-deficient cells (*Figure 7F*). Accompanying the increased stability of these substrates, we found other evidence to support premature S phase entry in SENP8-deficient cells, such as increased levels of the E2F targets Cyclin E1 and Cyclin A2, and decreased accumulation of the G1 cyclin Cyclin D1 and the CDK inhibitor p27 (*Resnitzky et al., 1994*; *Soucek et al., 1997*; *Sigl et al., 2009*; *Yuan et al., 2014*) (*Figure 7F*). Thus, we speculate that the increased stabilization of CRL substrates such as Cdc25A, UHRF1, and probably several other factors underlie the accelerated S phase entry and loss of proper G1/S phase regulation in SENP8-deficient cells.

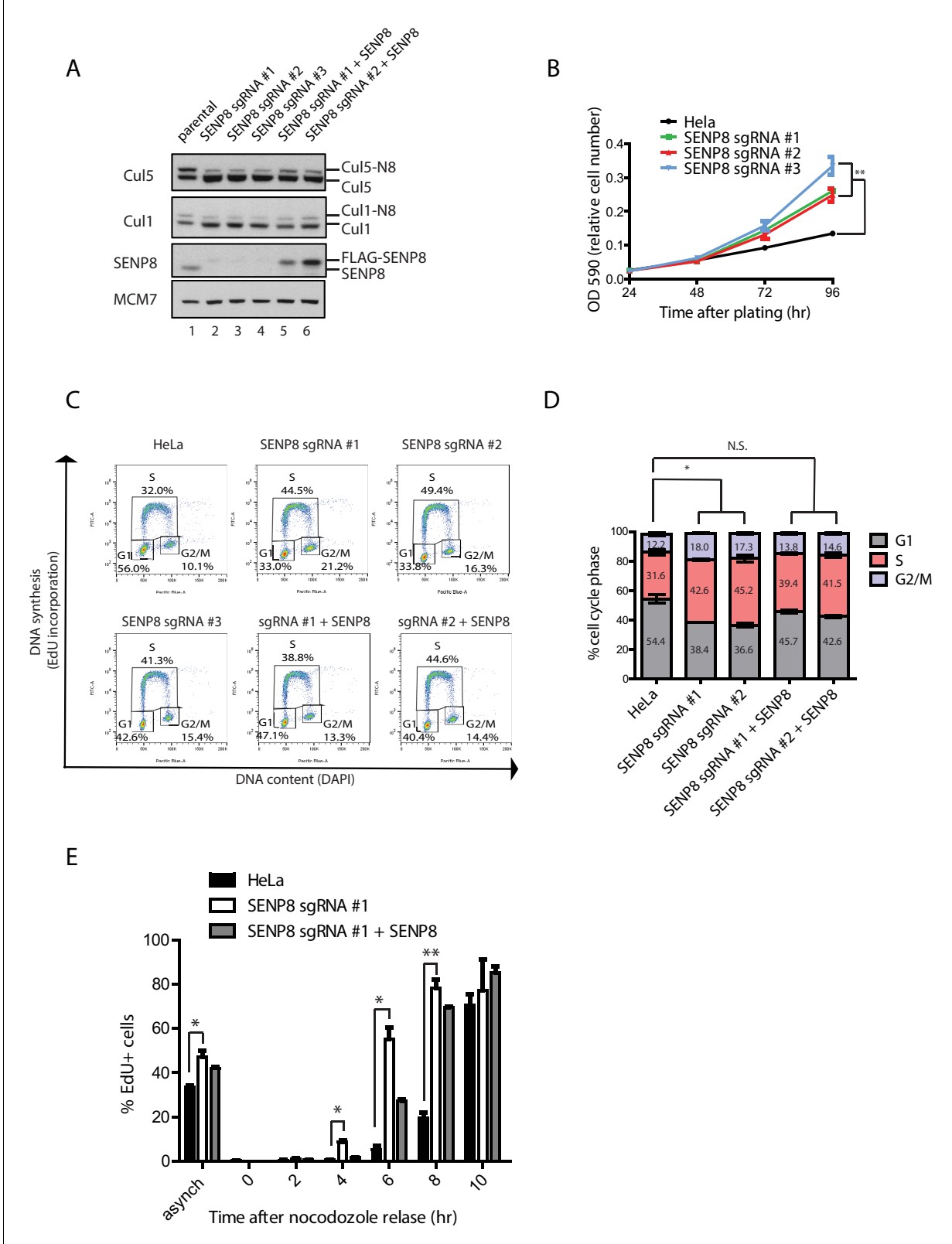

**Figure 6.** SENP8 depletion contributes to altered cell cycle progression and cell growth. (**A**) SENP8 knockout (lanes 2–4) and SENP8-rescued (lanes 5–6) HeLa cell lines were generated as specified in Materials and methods. Whole-cell lysates were analyzed by immunoblotting with the indicated antibodies. (**B**) Parental and SENP8 knockout HeLa cells were plated and stained with crystal violet at the indicated times. OD590 was measured to calculate relative cell numbers at each time point for triplicate samples. Error bars represent SEM. p Values were calculated for the 96 hr time point by

*Figure 6 continued on next page*

*Figure 6 continued*

paired Student's t-test. For all figures, asterisks represent either *$p<0.05$ or **$p<0.01$. (**C and D**) Parental and SENP8 knockout HeLa cells were pulse-labeled with 10 µM EdU for 30 min and analyzed by flow cytometry for DNA synthesis (EdU) and DAPI staining for DNA content (10,000 events per sample). Percentages of cells in G1, S, and G2/M phases are graphed for three biological replicates. Error bars indicate SEM. p Values were calculated for G1 populations by paired Student's t test. (**E**) Parental, SENP8-deficient, and SENP8-rescued HeLa cells were synchronized in prometaphase by thymidine-nocodozole block, collected by mitotic shake-off, and released into fresh media for the indicated times. Percentages of EdU-positive cells from three biological replicates were analyzed at each time point following nocodozole release. Statistical significance was calculated using a paired Student's t test. Error bars indicate SEM.

## Discussion

In this study, we developed a novel strategy to identify new non-cullin neddylation targets, adding to our current understanding of cellular roles for NEDD8 conjugation. Using a deconjugation-resistant FLAG-NEDD8-L73P mutant, we were able to isolate several NEDD8-conjugated proteins in cells for identification by MS (*Figure 1*), several of which were also found in previous analyses using ectopically-expressed NEDD8 (*Jones et al., 2008*; *Xirodimas et al., 2008*). Our system provides an advantage over previous proteomic approaches in that expression of even relatively low levels of the FLAG-NEDD8-L73P is sufficient to enable stabilization and IP of neddylated proteins for MS analysis, thus avoiding potential artificial effects of NEDD8 overexpression (*Xirodimas et al., 2008*; *Kim et al., 2011*; *Hjerpe et al., 2012*). This feature aided our detection of several NEDD8-conjugates unique to this study, including MCM7, DDB1, Rab14, FEM1C, RNF187 and others (see *Figure 1—source data 1* for complete list). Future work will be needed to further validate these as true neddylation substrates and functionally characterize these newly identified NEDD-conjugates in cells. Nevertheless, our MS dataset provides a valuable resource to expand upon previous studies of the NEDD8-modified proteome. Recently, several criteria for validation of *bona fide* neddylation substrates have been delineated (*Enchev et al., 2014*). In light of our current findings, we propose that identifying the physiological deneddylase could be added to these criteria to show potential dynamic and reversible neddylation of target substrates.

We identified and characterized the NEDD8 E2 conjugating enzyme Ubc12 as a novel substrate for auto-neddylation both in vitro and in cells (*Figure 2*). Through a combination of both MS-based analysis and Ubc12 Lys site mutagenesis, we identified the site of Ubc12 NEDD8 conjugation at its extreme N-terminus (K3). This modification is distinct from the site of Ubc12~NEDD8 thioester intermediate formation, occurring when NEDD8 is transferred from the active site of the E1 to the E2 active site Cys residue during the NEDD8 conjugation reaction (*Huang et al., 2007*). We have also shown that Ubc12 neddylation is sensitive to MLN4924 treatment and is dependent on the catalytic Cys residue of Ubc12. NEDD8 modification at the N-terminus of Ubc12 contributes to reduced DCN1 binding and consequently altered CRL activities, although the mechanism by which this occurs is still unclear. One possibility is that the NEDD8 modification could directly impede the initial interaction between the N-terminally acetylated Ubc12 and the hydrophobic binding pocket of DCN1 (*Scott et al., 2011*). Alternatively, NEDD8 modification could occur on pre-formed Dcn1-Ubc12 complex to induce a conformational change, leading to dissolution of the complex. Further in vitro analyses using purified components will be required to discriminate between these possibilities. Unexpectedly, from our SILAC-MS analysis (*Figure 4*), we identified additional components within the NEDD8 pathway that can form NEDD8-conjugates, including the E1 subunits UBA3 (UBE1C) and NAE1 (APPBP1), as well as the DCNLs DCUN1D1 and DCUN1D5. Future structural and biochemical work will be necessary to better understand the functional roles of these individual NEDD8 modifications in regulating the efficiency of NEDD8 conjugation to cullin and non-cullin substrates.

Furthermore, we identified the NEDD8-specific cysteine protease, SENP8, as the deneddylase that deconjugates NEDD8 from Ubc12 and other NEDD8 pathway components in human cells. Generation of CRISPR-derived knockout cell lines allowed us to profile effects of SENP8 depletion on global neddylation changes through MS-based proteomics (*Figure 4*) and immunoblot analysis of NEDD8 conjugates (*Figure 5*). These analyses revealed that SENP8 deneddylating activity is important to inhibit the aberrant neddylation of Ubc12 and other NEDD8 pathway components in cells (*Figures 4* and *5*). We also uncovered several previously uncharacterized non-cullin neddylation

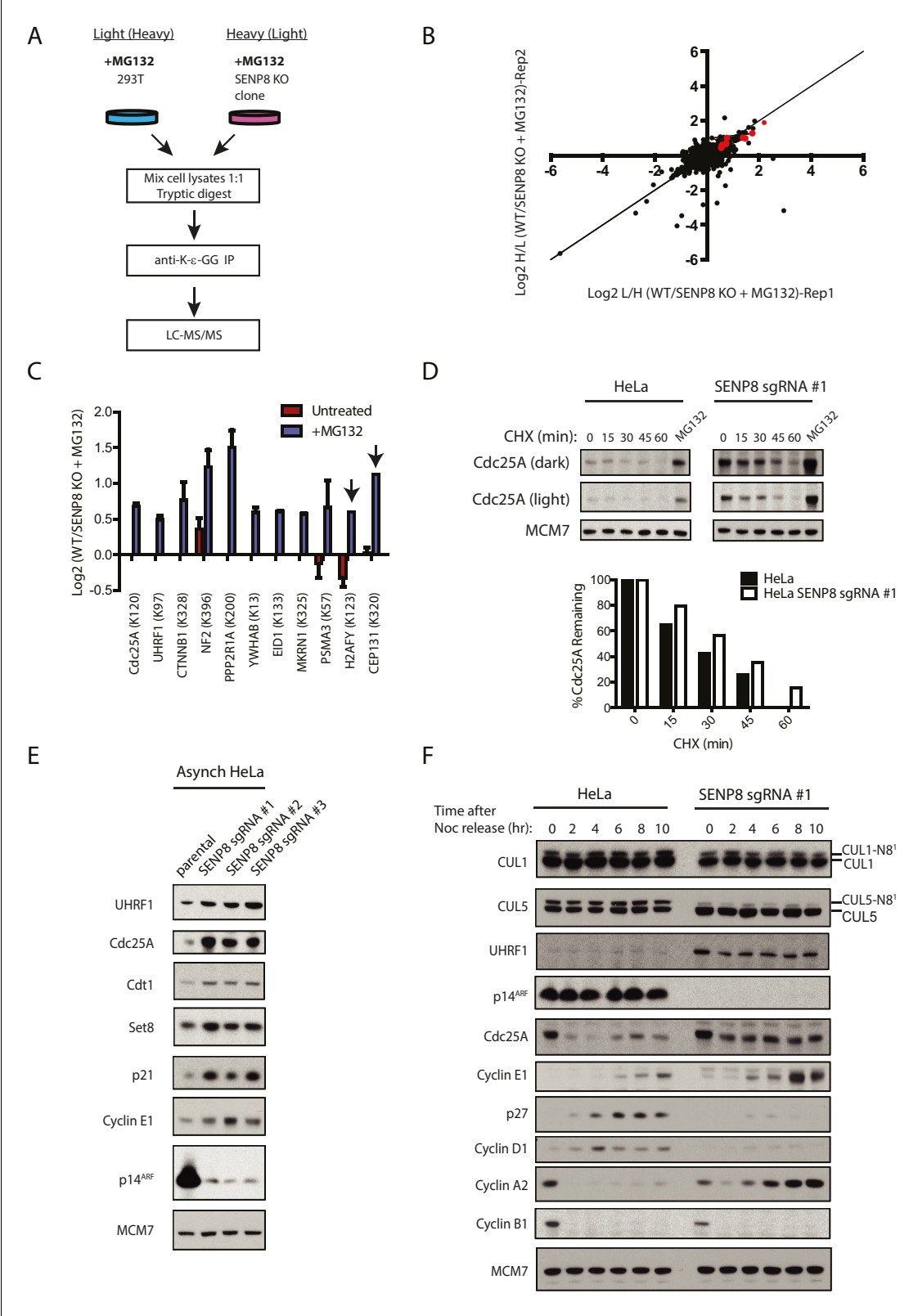

**Figure 7.** Loss of SENP8 contributes to decreased ubiquitylation and increased stability of CRL substrates. (**A**) Workflow for identification of K-ε-GG-modified peptides in MG132-treated WT and SENP8 knockout cells by antibody-based enrichment and quantitative SILAC-MS. Light- and Heavy-labeled HEK293T cells were treated with 10 μM of the proteasome inhibitor MG132 for 2 hr prior to harvesting and further processing. K-ε-GG-remnant-containing peptides of interest in (**C**) are highlighted in red. (**C**) Scatter plot showing Log2 ratios of all K-ε-GG sites identified in both the

*Figure 7 continued on next page*

*Figure 7 continued*

forward and reverse WT: SENP8 KO MG132-treated SILAC samples, prepared as in A. Line of identity with slope = 1 is shown for reference. (D) Log2 ratios of selected K-ε-GG modification sites in untreated versus MG132-treated SILAC samples. Absence of bars indicates that a particular K-ε-GG modification site was not detected. K-ε-GG sites that were detected in only one of the two replicate samples are indicated by arrows. Error bars represent SEM. (C) Parental and SENP8 knockout HeLa cells were treated with 30 µg/ml cycloheximide (CHX) for the indicated times and subjected to immunoblot analysis for Cdc25A levels. As a control, cell lines were treated with 10 µM of the proteasome inhibitor MG132 for 1.5 hr. Percentages of Cdc25A remaining were quantified in the parental and CRISPR knockout cell lines by densitometry analysis using ImageJ software. (D) Relative levels of CRL substrates were analyzed in lysates from parental and SENP8-deficient HeLa cells by immunoblotting analysis. (E) Parental and SENP8 knockout HeLa cells were synchronized in prometaphase by sequential thymidine and nocodozole blocks, released into fresh media, and collected at the specified times. Cell lysates were subjected to immunoblotting analysis for the indicated proteins.

The following source data and figure supplements are available for figure 7:

**Source data 1.** SILAC analysis of K-ε-GG remnant-containing peptides detected in MG132-treated parental and SENP8 knockout cell lysates.
**Figure supplement 1.** Statistical information related to K-ε-GG MS screen comparing MG132-treated parental and SENP8 knockout cell lysates (*Figure 7B*).
**Figure supplement 2.** SENP8-deficient cells show increased stability and reduced ubiquitylation of CRL substrates.

substrates in the SENP8 knockout cells (see *Figure 4—source data 1* for complete list), complementing our MS analyses using L73P-NEDD8 in *Figure 1F*.

Several adverse phenotypic consequences occur in SENP8 knockout cells, including decreased efficiency of NEDD8 conjugation to cullins (primarily CUL1 and CUL5), decreased CRL activity, and increased stabilization of CRL substrates, summarized in *Figure 8*. Accompanying these deficiencies, SENP8 knockout cells also showed aberrant G1/S cell cycle progression and accelerated cell growth (*Figure 6*). Interestingly, these cell cycle alterations partially phenocopy previously reported defects associated with the loss of either Cul1 (*Dealy et al., 1999*; *Wang et al., 1999*; *Chen and Li, 2010*) or Cul5 function (*Burnatowska-Hledin et al., 2001*; *Buchwalter et al., 2008*; *Bradley et al., 2010*;

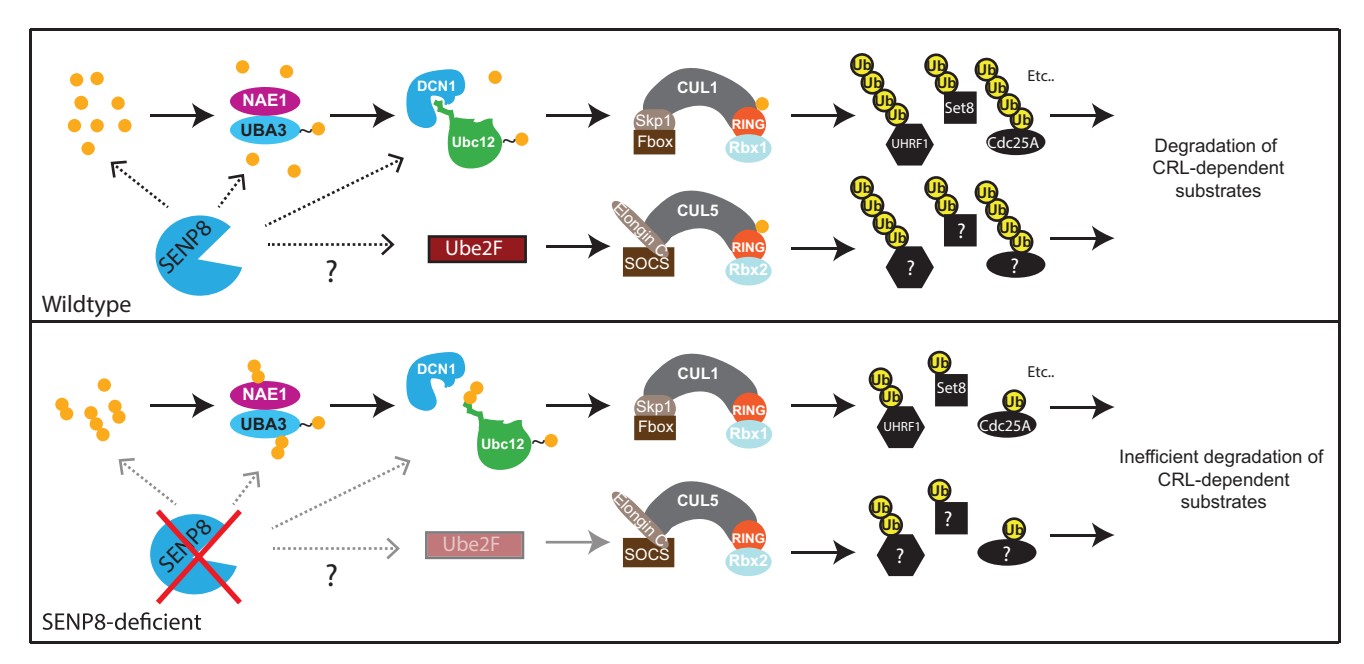

**Figure 8.** Model. A model depicting the role of SENP8 in regulating reversible neddylation of NEDD8 conjugation pathway components.

*Ma et al., 2013a*; *Willis et al., 2017*). Through an unbiased proteomic screen in which we enriched for K-ε-GG remnant-containing peptides in MG132-treated cells, we were able to identify several candidate substrates with deregulated ubiquitylation status and stability in the SENP8 knockout cells that could potentially contribute to G1/S progression defects. One prominent example is the SCF$^{\beta TrCP}$ substrate Cdc25A. Cdc25A acts as a major regulator of both the G1/S and G2/M cell cycle transitions by dephosphorylating and activating Cyclin E/Cdk2 and Cyclin B/Cdk1 complexes, respectively (*Strausfeld et al., 1991*; *Blomberg and Hoffmann, 1999*). Several reports have observed shortening of G1 phase and increased cyclin E- and cyclin A-associated kinase activity following overexpression of Cdc25A (*Blomberg and Hoffmann, 1999*; *Sexl et al., 1999*). Therefore, the increased total levels (*Figure 7E*) and impaired degradation of Cdc25A during interphase (*Figure 7F*) observed in SENP8 knockout cells could potentially play a significant role in promoting the unscheduled dephosphorylation of CDK complexes and accelerated cell cycle progression phenotypes.

Other SCF$^{\beta TrCP}$ substrates with differential ubiquitylation status in SENP8 knockout cells identified through our SILAC-MS screen (*Figure 7B*) were $\beta$-catenin and UHRF1. Aberrant ubiquitylation and degradation of $\beta$-catenin in SENP8-deficient cells may have a significant influence on cell growth by affecting transcription of downstream Wnt target genes such as c-Myc and Cyclin D1 (*He et al., 1998*; *Clevers and Nusse, 2012*). UHRF1 (ICBP90), an E2F1 target, also plays a critical role in controlling the G1/S transition by methylating and suppressing certain tumor suppressor genes (*Unoki et al., 2004*; *Jeanblanc et al., 2005*; *Alhosin et al., 2011*; *Taylor et al., 2013*; *Mudbhary et al., 2014*). Interestingly, UHRF1 has been shown to bind to the methylated promoter region of the p14$^{ARF}$ tumor suppressor (*Unoki et al., 2004*), which we also found was significantly downregulated in SENP8-deficient cells (*Figure 7E and F*). Whether the elevated UHRF1 levels in SENP8 knockout cells account for this reduced expression of p14$^{ARF}$ is unclear. However, deregulated p14$^{ARF}$ levels may have a significant impact on the accelerated G1/S progression in SENP8 knockout cells, as p14$^{ARF}$ plays a critical role as a tumor suppressor protein by blocking mouse double minute (MDM2)-induced degradation of p53 (*Stott et al., 1998*), and inhibiting E2F transcriptional activity (*Eymin et al., 2001*). The combined effects of p14$^{ARF}$ levels, deregulated CRL substrates identified in our SILAC-MS screen (*Figure 7B*), and undiscovered substrates (including unknown CUL5 targets, which are largely uncharacterized) are likely to contribute to the deregulated cell cycle progression observed in SENP8-deficient cells.

It is also apparent from our studies that the decreased CRL function accompanying SENP8 loss does not affect all CRL substrates uniformly and is likely to have substrate-specific effects. For example, SENP8 depletion does not entirely phenocopy the effects of complete inhibition of CRL activity following MLN4924 treatment, as we did not see a significant increase in cells with >4C DNA content (indicative of cells undergoing multiple rounds of replication without cell division, or re-replication) or induction of a DNA damage response in SENP8 knockout cells (data not shown). Similarly, using a CSN5-specific inhibitor developed by Novartis, it was recently shown that CSN inhibition does not have pleiotropic effects on all CRL substrates as MLN4924 treatment does (*Schlierf et al., 2016*), but results in substrate-specific stabilization. This effect was shown to be dependent on the auto-ubiquitylation of unique substrate recognition modules of CRLs following sustained neddylation and activation of CRL complexes. It is likely that SENP8 depletion has similar substrate-specific effects related to the dynamics of CRL activation and substrate-adapter auto-ubiquitylation influenced via aberrant neddylation of Ubc12.

An outstanding question from our study is concerning the biological role for auto-neddylation of NEDD8 pathway components: Is auto-neddylation simply a consequence of prolonged SENP8 depletion, or does it also occur in physiological conditions as a feedback mechanism regulating overstimulation of the NEDD8 pathway? Several recent studies have observed a correlation between overactivation of NEDD8 pathway components, including NEDD8, UBA3, NAE1, and Ubc12, with cancer progression and poor prognosis in human lung cancers (*Li et al., 2014*), colorectal cancers (*Xie et al., 2014*), and glioblastomas (*Hua et al., 2015*). Thus, auto-neddylation within the NEDD8 pathway may represent a deliberate quality control mechanism to limit adverse consequences associated with elevated NEDD8 conjugation and CRL activity. Consistent with this notion, we observed in our own study that NEDD8 conjugation to Ubc12 and other components occurs at defined Lys residues, which are evolutionarily conserved across species. If auto-neddylation of NEDD8 pathway components does in fact occur upon hyper-activation of the neddylation machinery in vivo, this also

raises the question of how SENP8 deneddylating activity is regulated under such circumstances. Furthermore, what defines the unique substrate preferences of SENP8 versus those of CSN, the other major deneddylating enzyme in human cells? These issues and others will require further interrogation of SENP8 regulation to resolve. Nevertheless, our study characterizes the SENP8 protease as a key regulator of the NEDD8 pathway, implicating SENP8 as a potential pharmacological target to control CRL-mediated processes in cells.

## Materials and methods

### Cell culture and lysis conditions

Both HeLa (ATCC CCL-2) and HEK293T (ATCC CRL-3216) cells were originally obtained from ATCC cell bank. These cell banks, including ATCC, comprehensively perform authentication and quality control tests on all distributed lots of cell lines. These cell lines are not in the list of commonly misidentified cell lines maintained by the International Cell Line Authentication Committee. The identity of these cell lines have been confirmed via STR profiling. All cell lines were periodically assessed for mycoplasma contamination using the Roche MycoTOOL Detection Kit. For all of our experiments, low-passage number of around 5–10 was used. HEK293T, HeLa, and HeLa Trex Flp-In cells (a gift from P. Jallepalli) were routinely maintained in 10% FBS in DMEM media, supplemented with L-glutamine. Transfections were carried out using Fugene 6 (Promega) or Hiperfect (QIAGEN), for plasmids or siRNA oligonucleotides, respectively, according to the manufacturer's protocol. Cells were routinely harvested in PBS, and the pellets were frozen at $-80°C$ prior to lysis. Cells were lysed in either denaturing SDS buffer (100 mM Tris [pH 6.8], 2% SDS and 20 mM $\beta$-Me) for direct analysis by SDS-PAGE or in nondenaturing buffer (50 mM Tris [pH 7.6], 150 mM NaCl, 2 mM EDTA, 0.5% NP-40, and protease inhibitor cocktail (Roche), and benzonase (Novagen) for IPs. For MLN4924 experiments, cells were treated with 5 μM MLN494 for 4 hr prior to harvesting. For cycloheximide chase experiments, cells were treated with 30 μg/ml cycloheximide or 10 μM MG132 for the indicated times before to harvesting in SDS lysis buffer. HeLa cells were synchronized in prometaphase by treatment with 2 mM thymidine for 18 hr followed by release into 100 ng/ml nocodozole for 10 hr. SILAC-labeled HEK293T cells were cultured in SILAC-heavy and light media for 10-doublings following protocols by the manufacturer (Pierce).

### Antibodies, plasmids, and siRNAs

Antibodies were purchased from the following sources for IP and/or immunoblotting: CUL1, −2,–3, −4A, −5, and Cdc25A (Bethyl), anti-FLAG M2 (Sigma), Ube2M (Ubc12), Ube1C (UBA3), APPBP1 (NAE1), DCUN1D1 (DCN1), Jab1 (CSN5), Rbx1, p21, $\alpha$-tubulin, Cyclin E1, Cyclin B1, Cyclin A2, and p14$^{ARF}$ (Abcam), HA (Covance), p27 (BD), Cyclin D1, SENP8, Ube2F, Cdt1, MCM7, and c-Myc (Santa Cruz Biotechnologies) NEDD8, Set8, and UHRF1 (Cell Signaling Technologies), and DCUN1D5 (Proteintech). HA-tagged Ub expression constructs were described previously (*Békés et al., 2013*). HA-tagged NEDD8 constructs were generated by PCR and cloned into pcDNA3.1. FLAG-SENP8 plasmid was purchased from Addgene (#18066). The retroviral vector QCXP-FLAG-SENP8 was engineered by recombinational cloning (Gateway LR clonase, Invitrogen) between QCXIP CMV TO/DEST (Addgene 17386) and pDONR221 plasmid containing a FLAG-SENP8 PCR product insert (generated using FLAG-SENP8 plasmid as template). Retroviral packaging was performed using standard protocols in HEK293T Phoenix cells followed by infection and selection in HeLa cells using 1 μg/mL puromycin. Ubc12 plasmids were purchased from OriGene and mutants were generated using the QuickChange Kit (Stratagene). All plasmids used in this study have been verified by sequencing. The siRNAs were synthesized by Qiagen and used at 20 μM final concentration: All-star Negative Control siRNA, SENP8 #1 (CCACTGGAGTTTATTGGTCTA), SENP8 #2 (ACCAACTTATTTGAACATTTA), and CSN5 (AAGAACAATATCCGCAGGGAA).

### Doxycyline-inducible FLAG-NEDD8 stable HeLa cell lines and CRISPR-mediated SENP8 knockout cells

Flp-In-FLAG-NEDD8 HeLa cells were generated as described previously using Flippase (Flp) recombination target (FRT)/Flp-mediated recombination technology in HeLa-T-rex Flp-In cells (*Tighe et al., 2008*; *Békés et al., 2013*), and expression of FLAG-Nedd8 was induced with 1 μg/mL doxycycline

for 24–48 hr. HeLa and HEK293T cells were co-transfected with Cas9 (Addgene #41815) and the gRNA-cloning vector (Addgene #41824) containing the following sgRNA target sequences against SENP8 coding sequences: sgRNA#1 – GGTAGCAGAGAAACTGG; sgRNA#2 – GTCATAGCTG TTTTGTT; sgRNA#3 – GACCAATAAACTCCAGT; sgRNA#4 – GTTGGAGTTATCATTGA (assembled by Gibson cloning) and transfected according the protocol described by Mali and Church et al. (*Mali et al., 2013*). SENP8 sgRNAs targeting the SENP8 coding region were designed by ZitFit tool (http://zifit.partners.org). Following transfection for 48 hr, cells were plated at 0.5 cell/well in 96-well dishes and cultured for 3 weeks. Individual outgrown clones were expanded and SENP8 knockout cells were verified by Western blotting. Out of ~15 SENP8 KO lines per cell line, three were chosen for further study per cell lines (data not shown).

## Expression and purification of recombinant proteins

For the purification of His-NEDD8-WT and -L73P constructs, NEDD8 was cloned into pET28b with an N-terminal His-tag. Briefly, *E. coli* codon-plus cells were transformed with the above constructs, and recombinant protein expression was induced with 1 mM IPTG at 30°C for 3 hr. His-tagged proteins were purified by $Ni^{2+}$-NTA followed by size exclusion chromatography on a Superdex-200 26/60 column in 20 mM Tris (pH 8.0), 350 mM NaCl, and 1 mM $\beta$-Me. His-tagged Ubc12 was produced in *E. coli* as previously (*Huang et al., 2009c*). His-SENP8 plasmid was a gift from Guy Salvesen and was purified as described previously (*Mikolajczyk et al., 2007*). Untagged Ubc12 and NAE1 were purchased from Boston Biochem. CUL1/Rbx1 and CUL3/Rbx1 were purchased from Ubiquigent. Purified COP9 signalosome was purchased from Enzo LifeScience. All constructs have been verified by sequencing and all recombinant proteins have been aliquoted and stored at −80°C until use.

## MS analysis of FLAG-NEDD8 immunoprecipitations

Cell pellets from an equal number of cells were lysed in mRIPA buffer (20 mM Tris [pH 7.5], 1% NP-40, 0.5% CHAPS, 0.1% SDS, and 150 mM NaCl) supplemented with protease inhibitor cocktail (Roche) and 1 U/µl benzoase (Novagen), for 1 hr on ice. Lysates were cleared by centrifugation and the supernatant was incubated 100 µg M2-conjugated (anti-FLAG) magnetic DynaBeads (Sigma-Aldrich) for 1 hr at 4°C. The beads were washed in mRIPA buffer and in 100 mM ammonium bicarbonate (pH 8). FLAG-NEDD8 immunoprecipitation samples were reduced with 0.2 M dithiothreitol (DTT) at pH 8 for 1 hr at 57°C and subsequently alkylated using 0.5 M iodoacetamide at pH 8 for 45 min in the dark at RT. Then, 200 ng trypsin (Promega) was added to the samples still bound to the anti-FLAG beads. Digestion was allowed to proceed overnight with gentle shaking at room temperature. The resulting peptide mixture was removed from the beads and K-ε-GG containing peptides were enriched with the PTMScan Ubiquitin Remnant Motif (K-ε-GG) antibody (Cell Signaling Technology) following the kit instructions. Briefly, peptide samples were added to two vials of washed beads and the mixture was rotated at 4°C for 2 hr. Samples were centrifuged at 2000 x g for 30 s at 4°C, the supernatant was removed, and the beads were washed twice with IAP buffer, and three times with water. Bound peptides were eluted with three rounds of 10 mM citric acid. Pooled elutions and supernatants were acidified with trifluoroacetic acid, concentrated in a SpeedVac concentrator and desalted as previously described (*Cotto-Rios et al., 2012*).

An aliquot of each IP supernatant and elution sample was loaded onto an Acclaim PepMap trap column in line with an EASY-Spray 50 cm x 75 µm ID PepMap C18 analytical HPLC column with 2 µm bead size using the auto sampler of an EASY-nLC 1000 HPLC (ThermoFisher) and solvent A (2% acetonitrile, 0.5% acetic acid). The peptides were gradient eluted into a Q Exactive (Thermo Scientific) mass spectrometer using a 120 min gradient from 2% to 40% solvent B (90% acetonitrile, 0.5% acetic acid), followed by a 10 min ramp up to 100% solvent B and was held at 100% for 10 min.

High-resolution full MS spectra were acquired with a resolution of 70,000 (@ m/z 200), an AGC target of 1e6, with a maximum ion time of 120 ms, and scan range of 400 to 1500 m/z. Following each MS1, 20 data-dependent high-resolution HCD MS2 spectra were acquired. All MS2 spectra were collected of precursors of charge states 2–5 using the following instrument parameters: resolution of 17,500, AGC target of 5e4, maximum ion time of 250 ms, one microscan, 2 m/z isolation window, 30 s dynamic exclusion, and Normalized Collision Energy (NCE) of 27.

All acquired MS2 spectra were searched against a human uniprot database using Sequest within Proteome Discoverer (Thermo Scientific). In the search parameters, trypsin was selected with two

missed cleavages, precursor mass tolerance was set to ±10 ppm, and fragment ion mass tolerance was set to ±0.02 Da. Carboxymethylation of Cys was added as a static modification. The following variable modifications were allowed: oxidation of methionine and deamidation of glutamine and asparagines, N-terminal acetylation, and lysine K-ε-GG. All results were filtered to only include peptides identified with high confidence and proteins identified by at least two peptides. Due to the low efficiency of the K-ε-GG antibody IP a number of K-ε-GG modified peptides were identified in the flow through. Results from the mass spectrometric analysis of the K-ε-GG IP flow through and elution were combined and peptide spectral matches from the flow through and elution were summed to generate the data shown in *Figure 1F*.

## SILAC-MS analysis of K-ε-GG-modified peptides

Light and heavy SILAC K(13C(6))-labeled cells were lysed in 8M urea according to the manufacturer's protocol, quantified by BCA assay, and pooled 1:1. The mixed lysates were reduced with 0.2 M dithiothreitol (DTT) at pH 8 for 1 hr at 57°C and subsequently alkylated using 0.5 M iodoacetamide at pH 8 for 45 min in the dark at RT. Digestion with trypsin at a 1:100 ratio (Promega) was allowed to proceed overnight at room temperature. Peptide mixtures were desalted using a tC18 Sep-Pak Vac 3cc 200 mg capacity (Waters) and freeze dried using a lyophilizer (ScanVac).

Dried peptides were reconstituted in the provided IAP buffer and K-ε-GG containing peptides were enriched with the PTMScan Ubiquitin Remnant Motif (K-ε-GG) Kit (Cell Signaling Technology) following the kit instructions as described above.

An aliquot of the K-ε-GG IP inputs and elutions were loaded as described above. The peptides were gradient eluted into a Q Exactive (Thermo Scientific) mass spectrometer using a 180 min gradient from 2% to 20% solvent B (90% acetonitrile, 0.5% acetic acid), 20 min from 20–40% solvent B, followed by a 10 min ramp up to 100% solvent B and was held at 100% for 20 min. Acquisition was performed as described above.

Peptide and protein identification and SILAC quantitation was performed using the MaxQuant software suite version 1.5.2.8 (*Cox and Mann, 2008*) against Uniprot Human database. For the first search, the peptide mass tolerance was set to 20 ppm and for the main search peptide mass tolerance was set to 4.5 ppm. Trypsin-specific cleavage was selected with two missed cleavages. Both peptide spectral match and protein FDR were set to 1% for identification. The multiplicity was set to two, with three maximum labeled residues, and the heavy label of Lys6 selected. Carboxymethyl of cysteine was added as a static modification. Oxidation of methionine, deamidation of glutamine and asparagine, acetylation of N-termini and GlyGly of lysine were allowed as variable modifications. Protein quantitation was performed using unique and razor peptides. Normalized SILAC ratios or the inverse were used to determine the relative abundance of K-ε-GG modified peptides in SENP8 Knockout cells as compared to WT.

## Flow cytometry analysis

Cells to be analyzed by flow cytometry for DNA content were pulse-labeled with 10 μM EdU prior to trypsinization and fixation in 4% paraformaldehyde. EdU incorporation was detected using the Alexa Fluor 488 Click-iT EdU Flow Cytometry assay kit (Thermo Fisher Scientific) and DNA was counterstained with 1 μg/ml DAPI (Molecular Probes) prior to analysis. Flow cytometry data acquisition was performed on an LSR II and data analysis was performed using FACS Diva software (BD Pharmingen, San Diego, CA) or FlowJo software (Treestar, Ashland, OR).

## In vitro Ubc12 auto-neddylation and deneddylation assays

For in vitro neddylation reactions, 1 μM NAE1, 2 μM Ubc12 and 15 μM Nedd8 were added to reactions containing 50 mM Tris, pH 7.6, 150 mM NaCl, 10 mM MgCl$_2$, 1 mM DTT and 5 mM ATP. Reactions were incubated at 37°C for 15 min, terminated with 4X LDS loading buffer supplemented with 50 mM DTT and analyzed by SDS-PAGE and SYPRO staining. For CUL/Rbx1 neddylation reactions, CUL1/Rbx1 was also added at 1 μM each and reactions were incubated for 30 min at 37°C (or 1 min at 37°C in the presence of DCN1). For deneddylation reactions, the initial neddylation reaction was quenched with 1 M DTT and 0.5 M EDTA, diluted twofold, and incubated for another 30 min at 37°C with purified CSN complex (0.5 μg).

## Cell growth assay

WT and SENP8-deficient HeLa cells were plated at a density of $1 \times 10^5$ cells/well of a 12-well dish and stained with crystal violet solution (0.5% crystal violet w/v, 25% methanol) at the following times after plating: 24, 48, 72, and 96 hr. OD590 was measured at each time point in triplicate to quantify relative cell numbers.

## Acknowledgements

The authors would like to thank Jie Sun and Talia Hart (NYU-SURP) for technical assistance, Brenda Schulman and Chris Lima for technical expertise and reagents on Ubc12 E2 auto-neddylation assays, and members of the Huang lab for technical assistance and critical reading of the manuscript. We also thank members of the Pagano laboratory for reagents and advice related to cell cycle and ubiquitylation assays. MB was supported by NIH NRSA fellowship (F32GM100598). Work in the TTH laboratory is supported by grants from the NIH (ES025166), ACS (RSG-12–158-DMC), and from the NYU Laura and Isaac Perlmutter Cancer Center Support Grant's Developmental Project Program P30 CA016087. Work in the laboratory of BU and flow cytometry facility (Peter Lopez) is partially supported by the Cancer Center Support Grant, P30CA016087, at the Laura and Isaac Perlmutter Cancer Center.

## Additional information

### Funding

| Funder | Grant reference number | Author |
|---|---|---|
| National Institutes of Health | F32GM100598 | Miklós Békés |
| National Cancer Institute | Cancer Center Support Grant P30CA016087 | Beatrix M Ueberheide Tony T Huang |
| National Institutes of Health | ES025166 | Tony T Huang |
| American Cancer Society | RSG-12-158-DMC | Tony T Huang |

The funders had no role in study design, data collection and interpretation, or the decision to submit the work for publication.

### Author contributions

KEC, Conceptualization, Data curation, Formal analysis, Investigation, Methodology, Writing—original draft, Writing—review and editing; MB, Conceptualization, Data curation, Formal analysis, Investigation, Methodology, Writing—review and editing; JRC, Resources, Data curation, Formal analysis, Methodology, Writing—review and editing; SBC, MJKJ, Data curation, Investigation; BMU, Formal analysis, Supervision, Funding acquisition, Resources; TTH, Formal analysis, Supervision, Funding acquisition, Writing—review and editing

### Author ORCIDs

Kate E Coleman, http://orcid.org/0000-0003-4676-4981
Tony T Huang, http://orcid.org/0000-0001-9291-5002

## Additional files

### Major datasets

The following dataset was generated:

| Author(s) | Year | Dataset title | Dataset URL | Database, license, and accessibility information |
|---|---|---|---|---|
| Coleman KE, Bekes M, Chapman JR, Crist SB, Jones MJ, | 2017 | SENP8 limits aberrant neddylation of NEDD8 pathway components to promote cullin-RING ubiquitin | http://proteomecentral. proteomexchange.org/ cgi/GetDataset?ID= | Publicly available at Proteome Xchange (accession no. |

| Ueberheide BM, Huang TT | ligase function | PXD006288 | PXD006288; MassIVE ID MSV000080917) |
|---|---|---|---|

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
