## [Decision Letter]

Thank you for submitting your article "SENP8 limits aberrant neddylation of NEDD8 pathway components to promote cullin-RING ubiquitin ligase function" for consideration by *eLife*. Your article has been reviewed by three peer reviewers, one of whom is a member of our Board of Reviewing Editors, and the evaluation has been overseen by Tony Hunter as the Senior Editor. The reviewers have opted to remain anonymous.

The reviewers have discussed the reviews with one another and the Reviewing Editor has drafted this decision to help you prepare a revised submission.

Summary:

Posttranslational modification with NEDD8 is required for activation of the large class of CULLIN-RING-ligases, and hence, it is essential for cell division and survival. As inhibitors of NEDD8 conjugation have entered clinical trials, it is important to understand the complete substrate spectrum of NEDDylation, as well as the regulation of this modification. Previously, several non-cullin NEDD8 substrates have been proposed; however, these studies have relied on overexpression of NEDD8, which can lead to non-specific protein modification through mis-targeting by ubiquitylation enzymes. Developing new methods to identify potential non-cullin NEDD8 substrates is, therefore, important.

In this manuscript Békés et al. developed a deconjugation-resistant form of NEDD8, and then exploited this protein to identify novel substrates that are modified with NEDD8 in the cell. Following this strategy, they successfully identified non-cullin neddylation targets including the NEDD8 E2, UBC12, which they showed undergoes neddylation on Lys3. They further showed that this modification interferes with the interaction of UBC12 with the co-E3 DCN1 thereby downregulating CUL neddylation. In parallel, they showed that SENP8 is the enzyme responsible for removing this modification from UBC12 as well from other substrates. Finally they demonstrated that SENP8 depletion contributes to accumulation of CRL substrates and defective cell cycle progression.

Most experiments are well-conceived and executed well. This work clearly demonstrates that components of the neddylation machinery can be neddylated themselves and it is likely that sifting through the supplemental tables will reveal new neddylation targets that were not characterized in this work. However, several issues need to be addressed before this manuscript can be accepted for publication.

Essential revisions:

1) Further characterization of the inhibitory function of Ubc12 neddylation is required to provide a solid foundation for the phenotypes observed in this work. For example, it is a clear prediction that SENP8 should activate CRL1 in in vitro ubiquitylation assays. As all proteins are available, the authors should test this notion. A positive result in this experiment would also support the broad claims of SENP8 function in the title.

2) The data suggesting that UBC12 neddylation inhibits the interaction of this protein with DCN1 is not particularly strong. The authors should purify neddylated UBC12 and test its ability to engage DCN1 in direct binding assays. Furthermore, they should re-introduce Lys3 into their NK0 mutant of UBC12 and test whether this is sufficient to disrupt interaction with DCN1 (the data concerning the K3R mutation are confusing, but this experiment should strengthen the notion that K3 neddylation has functional impact). A better description of the NK0 mutant in the text would also be helpful, including control experiments that demonstrate that the NK0 mutant of UBC12 does not interfere with acetylation of this protein.

3) The last figure, Figure 6, describing the in vivo relevance of this autoneddylation/SENP8 pathway is rather incomplete, and for publication in *eLife* stronger physiological data are required, to establish that the perturbation in cell cycle progression observed in SENP8 knockdown cells is in fact due to a failure to deneddylate UBC12 and the resultant decrease in CRL activity, as opposed to the lack of SENP8 deneddylation of some other protein involved in cell cycle progression. For example, the authors could investigate their mass spec data for global changes in CRL substrate levels or they could look at more candidate CRL substrates and investigate the consequences of their stabilization (it is unclear whether the tested ones are the only ones with a phenotype, which would raise some doubts about the importance of this pathway). Along these lines, does NEDD8-L73P produce similar phenotypes as deletion of SENP8?

4) A better statistical analysis and complete representation of the mass spec data is required. In particular, they need to clarify how many substrates were identified, how many were validated; what are real interactions and what is background?

---

## [Author Response]

*Essential revisions:*

*1) Further characterization of the inhibitory function of Ubc12 neddylation is required to provide a solid foundation for the phenotypes observed in this work. For example, it is a clear prediction that SENP8 should activate CRL1 in in vitro ubiquitylation assays. As all proteins are available, the authors should test this notion. A positive result in this experiment would also support the broad claims of SENP8 function in the title.*

We understand the importance of an in vitro reconstitution experiment to show the direct role of SENP8 in activating DCN1-dependent Cul1 ligase activity in a ubiquitylation assay, however, the setup for this experiment is not as straightforward as originally anticipated. For this assay to be executed properly, we envision multiple sequential reactions that would need to be each independently optimized. For example, we would need to demonstrate in an in vitro system that:

1) Recombinant SENP8 purified from either human cells or insect cells can directly cleave NEDD8 from aberrantly neddylated Ubc12 in the presence of charged E1;

2) Deneddylated Ubc12 is then fully functional to promote DCN1-dependent Cul1 neddylation;

3) In vitro neddylated Cul1 can be reconstituted in an active SCF E3 ubiquitin ligase complex and used to in vitro ubiquitylate a candidate substrate and lastly;

4) SCF-mediated substrate ubiquitylation occurs more efficiently (or not) with increasing concentrations of SENP8 in the original deneddylation reaction. This multi-step reaction would also need to be optimized for various reaction parameters, including concentrations of enzymes and substrates, and appropriate timing of in vitro neddylation and ubiquitylation reactions.

Unfortunately, we do not have the capacity nor the technical expertise at this stage to execute this complicated biochemical reconstitution experiment within the expected timeline for revision. One of the major challenges is our inability to purify a sufficient amount of mammalian Ubc12 WT and NK0 mutant from tissue culture cells to perform the DCN1-dependent Cul1 neddylation reaction optimally. We were able to use bacterially-expressed Ubc12 as a recombinant source to study the mechanism of Ubc12 auto-neddylation and direct deneddylation by recombinant SENP8 proteins (Figure 3, and Figure 3—figure supplement 2). However, for the in vitro reconstitution experiments described above, purification of Ubc12 from mammalian or insect cells would be necessary to preserve the integrity of the N-terminal acetylation mark on Ubc12, since the N-term acetylation would be lost by expression/purification in a bacterial system (N-terminal acetylation is essential for Ubc12 binding to DCN1).

Nevertheless, we attempted to address some aspects of this question using epitope- tagged Ubc12 IP-ed from human cells, although the yield of Ubc12 was relatively low for biochemical studies. Using this source of Ubc12, we found a major difference in the kinetics of autoneddylation of Ubc12 versus DCN1-dependent neddylation of Cul1 (data not shown). For example, while in vitro DCN1-dependent Cul1 neddylation can occur within a time-frame of around 1 minute, Ubc12 auto-neddylation is observed between 30-60 min. In other words, most Cul1 is already neddylated in vitro before Ubc12 auto- neddylation takes place. Therefore, we predicted that this would further complicate our ability to assess how aberrant Ubc12 auto-neddylation alters Cul1 neddylation efficiency in an in vitro assay, in which components for both Cul1 and Ubc12 neddylation would be mixed together simultaneously. Related to this point, we think that Ubc12 auto- neddylation is a relatively slow event and accumulates over time in the absence of SENP8. In support of this idea, we were able to see much more auto-neddylated Ubc12 accumulate in the CRISPR-knockout SENP8 clones in comparison to cells transiently depleted for SENP8 using siRNAs (compare Figure 3). So, the overall defect in CRL activity that we observed in the absence of SENP8 is probably a cumulative effect over several cell divisions, attributable to a gradual decrease in the pool of Ubc12 that is readily available to transfer E1-activated Nedd8 to Cul1 via DCN1 binding. In the future, we hope to directly address this point to understand both spatial and temporal regulation of Ubc12 aberrant neddylation.

To support the role of SENP8 in activating Cul1 in cells, we performed new experiments to show that 1) SENP8 knockout cells have less binding of endogenous Ubc12 to DCN1, which can be rescued by transiently expressing SENP8 WT in SENP8- deficient cells (new Figure 5—figure supplement 1); and that 2) SENP8-deficient cells contain a subset of Cul1 target proteins with reduced ubiquitylation in comparison to WT cells (identified using quantitative proteomics and candidate-based Western blot detection of ubiquitylated proteins) (new Figure 7 and Figure 7—figure supplement 2). This further supports our claim that SENP8 suppresses aberrant neddylation of components of the NEDD8 pathway to promote efficient CRL-dependent ubiquitylation and degradation of substrates in cells. Although the defect in CRL-dependent degradation of substrates in SENP8-deficient cells is attributable to aberrant Ubc12 auto-neddylation based on our indirect evidence of reduced DCN1 binding, we cannot rule out the possibility that other targets of aberrant neddylation in the conjugation pathway, such as DCN1, or Nae1-Uba3, or even the free Nedd8 pool itself, could synergistically contribute to the CRL activation defect phenotype we observed. We have mentioned this possibility in the Discussion section.

*2) The data suggesting that UBC12 neddylation inhibits the interaction of this protein with DCN1 is not particularly strong. The authors should purify neddylated UBC12 and test its ability to engage DCN1 in direct binding assays. Furthermore, they should re-introduce Lys3 into their NK0 mutant of UBC12 and test whether this is sufficient to disrupt interaction with DCN1 (the data concerning the K3R mutation are confusing, but this experiment should strengthen the notion that K3 neddylation has functional impact). A better description of the NK0 mutant in the text would also be helpful, including control experiments that demonstrate that the NK0 mutant of UBC12 does not interfere with acetylation of this protein.*

To address these points, we now include new data to show that WT Ubc12 interacts with relatively less endogenous DCN1 when WT Ubc12 is irreversibly auto- neddylated by HA-Nedd8-L73P (new Figure 2). We also performed new co- immunoprecipitation analyses to show that endogenous Ubc12 interacts with less DCN1 in SENP8 knockout cells in comparison to SENP8-corrected cells (new Figure 5—figure supplement 1). This further supports our conclusion that in conditions that promote Ubc12 auto-neddylation (loss of SENP8), there is a decrease in Ubc12 found associated with DCN1, leading to a reduction in Cul1 activation.

In hindsight, we realize how the K3R data can be somewhat confusing. Our original expectation was that this mutation would block Ubc12 auto-neddylation by HA- Nedd8-L73P, as the K3 site was identified as a K-ε-GG-modified residue by targeted MS analysis in Figure 2. However, we found that the K3R Ubc12 mutant could still support neddylation (likely due to the compensatory nature of the multiple Lys sites available at the N-terminal region of Ubc12), prompting us to create an additional mutant (NK0) to block all potential Lys sites for NEDD8 modification at the N-terminus. To eliminate this confusion, we decided to remove the K3R data from the paper, and instead generated the construct suggested above with K3 added back to the NK0 mutant of Ubc12 (N-K3). As shown in the new Figure 2, we found that the N-K3 mutant can partially rescue Ubc12 auto-neddylation by HA-Nedd8-L73P, in agreement with our MS results mapping K3 as an in vivo site for auto-neddylation. Unfortunately, we were unable to convincingly show that the N-K3 mutant can block interaction with DCN1 (data not shown). We think this is because the N-K3 mutant only partially rescues Ubc12 autoneddylation, and there is still a sufficient amount of unmodified Ubc12 available to interact with DCN1 in comparison with the NK0 mutant. In the future, we hope to develop better molecular strategies to purify the neddylated Ubc12 form away from unmodified Ubc12 to directly measure binding properties between DCN1 and Ubc12.

It was an interesting suggestion to determine whether the NK0 mutations impact N-terminal acetylation of Ubc12. It is widely assumed that N-terminal acetylation of translated proteins is a constitutive event and not subjected to regulation, but this is something that could indeed be interesting to explore further as a way to regulate DCN1 and Ubc12 binding. According to original IP analyses in Figure 2, the NK0 mutant of Ubc12 showed increased binding to DCN1 in comparison to WT Ubc12. Since N- terminal acetylation was shown to be a requirement for the interaction between Ubc12 and DCN1 (Scott et al., 2011), it follows that the NK0 Ubc12 mutant must also be acetylated; if this were not the case, then we would not have observed any association of the NK0 Ubc12 mutant with DCN1 by co-IP. By this argument, we have decided to forgo efforts to validate N-terminal acetylation of the NK0 mutant to focus attention on other suggested experiments.

*3) The last figure, Figure 6, describing the in vivo relevance of this autoneddylation/SENP8 pathway is rather incomplete, and for publication in eLife stronger physiological data are required, to establish that the perturbation in cell cycle progression observed in SENP8 knockdown cells is in fact due to a failure to deneddylate UBC12 and the resultant decrease in CRL activity, as opposed to the lack of SENP8 deneddylation of some other protein involved in cell cycle progression. For example, the authors could investigate their mass spec data for global changes in CRL substrate levels or they could look at more candidate CRL substrates and investigate the consequences of their stabilization (it is unclear whether the tested ones are the only ones with a phenotype, which would raise some doubts about the importance of this pathway). Along these lines, does NEDD8-L73P produce similar phenotypes as deletion of SENP8?*

To address these concerns, we performed the following new experiments:

We took a more unbiased approach to identify CRL target proteins that are less ubiquitylated (more stable) in the absence of SENP8. We utilized SILAC-MS (K-ε-GG) proteomics analysis of MG132-treated WT vs SENP8 knockout cells to identify additional relevant substrates that have compromised ubiquitylation status in SENP8 knockout cells. Some of the substrates that showed differential ubiquitylation status under MG132 treatment have key roles in controlling cell proliferation (new Figure 7). Moreover, we were able to verify the altered stability of these proteins, such as Cdc25A, in both asynchronous and G1/S-synchronized cells (new Figure 7), and in cycloheximide chase experiments (new Figure 7 and Figure 7—figure supplement 2). Although it is likely that proteins other than the ones tested by immunoblotting may contribute to the growth and cell cycle defects in SENP8-deficient cells, our K-ε-GG profiling MS screen adds substantial impact to our study by providing a broad overview of major proteins/pathways affected by SENP8 loss, many of which fall under the category of defective cullin neddylation.

We provide new evidence that overexpression of WT Ubc12, NK0 Ubc12, and WT DCN1 can partially rescue the defective neddylation of CUL1/CUL5 in SENP8 knockout cells (Figure 5—figure supplement 1). This strengthens the notion that failure to deneddylate Ubc12 leads to suboptimal NEDD8 pathway function in neddylating and activating CRLs, since adding back excess Ubc12 and DCN1 can partially rescue the cullin neddylation defects observed in SENP8 knockout cells. We also provide new data showing that complementation of SENP8 knockout HeLa cells with WT SENP8 partially rescues the effects of SENP8 loss on G1/S distribution and cullin neddylation (new Figure 6). Additionally, to more thoroughly characterize the G1/S defect observed in SENP8-deficient cells, we performed a synchronization experiment to compare the timing of G1 to S phase entry in WT versus SENP8 knockout HeLa cells. Using combined flow cytometry analysis of EdU incorporation (new Figure 6) and immunoblotting analyses of various cell cycle regulators (new Figure 6), we showed that SENP8-deficient cells enter S phase prematurely in comparison to WT cells.

Furthermore, complementation of SENP8 knockout cells with WT SENP8 partially reverses this accelerated G1-S progression (new Figure 6).

In summary, the new data provides stronger physiological evidence to explain how the loss of SENP8 and consequent effects on aberrant Ubc12 auto-neddylation can lead to a G1-S progression defect. We do not think the Nedd8-L73P would produce a similar phenotype as the deletion of SENP8 since the Nedd8-L73P mutant also stabilizes cullin neddylation levels. Therefore, the downstream functional defect of Nedd8-L73P expression would likely be attributable to combined phenotypes associated with hyper-neddylation of Ubc12 and cullin proteins (aberrant Ubc12 neddylation equates to less CRL activity, while hyper-neddylation of CRL could provide more CRL activity, but less F-box turnover). Therefore, we did not attempt to measure the downstream effects of Nedd8-L73P expression because the interpretation of the data would be confusing due to the pleiotropic effects of this Nedd8 mutant.

*4) A better statistical analysis and complete representation of the mass spec data is required. In particular, they need to clarify how many substrates were identified, how many were validated; what are real interactions and what is background?*

To provide a statistical overview of the SILAC-MS datasets featured in Figure 4 and new Figure 7, we have included additional supplemental figures (new Figure 4—figure supplement 1 and Figure 7—figure supplement 1) with information regarding numbers of peptides, proteins, PSMs, and K-ε-GG sites identified in each experimental replicate. Venn diagrams included in these same supplemental figures show the degree of overlap between all K-ε-GG-modified peptides identified between replicates and between untreated and MG132-treated samples. All attempts to validate neddylation targets were done by immunoblotting and presented in Figure 5 and Figure 7. Gene ontology (GO) term analysis is provided for Figure 4 and Figure 7 to show the major biological processes and molecular functions of candidate neddylation and ubiquitylation substrates identified in each screen (new Figure 4—figure supplement 1 and Figure 7—figure supplement 1).